# Identifying Metric Structures of Deep Latent Variable Models

**Stas Syrota** [1]   **Yevgen Zainchkovskyy** [1]   **Johnny Xi** [2]   **Benjamin Bloem-Reddy** [2]   **Søren Hauberg** [1]

## Abstract

Deep latent variable models learn condensed representations of data that, hopefully, reflect the inner workings of the studied phenomena. Unfortunately, these latent representations are not statistically identifiable, meaning they cannot be uniquely determined. Domain experts, therefore, need to tread carefully when interpreting these. Current solutions limit the lack of identifiability through additional constraints on the latent variable model, e.g. by requiring labeled training data, or by restricting the expressivity of the model. We change the goal: instead of identifying the latent variables, we identify *relationships* between them such as meaningful distances, angles, and volumes. We prove this is feasible under very mild model conditions and without additional labeled data. We empirically demonstrate that our theory results in more reliable latent distances, offering a principled path forward in extracting trustworthy conclusions from deep latent variable models.

## 1. Introduction

Latent variable models express the density of observational data through a set of latent, i.e. unobserved, variables that ideally capture the driving mechanisms of the data-generating phenomena. For example, the latent variables of a variational autoencoder (Kingma & Welling, 2013; Rezende et al., 2014) trained on protein data, can reveal the underlying protein evolution which can help domain experts understand a problem of study (Riesselman et al., 2018; Ding et al., 2019; Detlefsen et al., 2022).

Unfortunately, latent variables are rarely identifiable, i.e. they cannot be uniquely estimated from data. This lack of uniqueness prevents reliable analysis of the learned latent

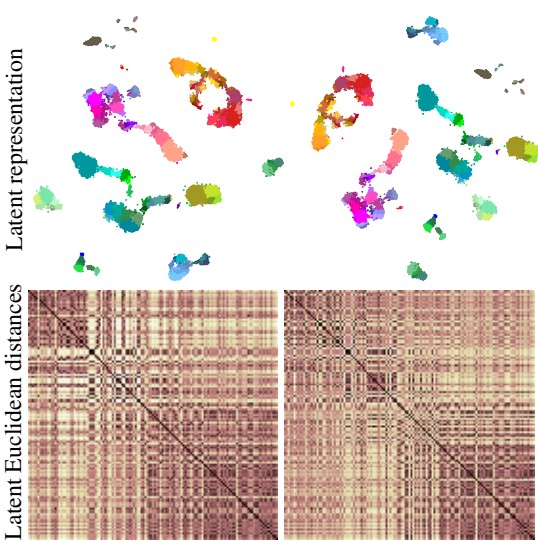

Figure 1: Latent representations of transcriptomic data (top row) changes with model retraining. Each column corresponds to a model trained from scratch. The latent variables are not identifiable and change between training runs. Pairwise Euclidean distances (bottom row, averaged across cell types) also change significantly between runs. This lack of identifiability prevents us from reliably using the latent representations to understand the underlying biology.

variables as the analysis becomes subject to the arbitrariness of model training. Fig. 1 exemplifies the issue, where two independently trained latent representations are estimated from transcriptomic data (Tasic et al., 2018). While clusters are similarly estimated across models, the *relationships* between clusters vary significantly, preventing us from extracting intra-cluster information from the plots. This fundamental issue has sparked the development of training heuristics to limit the issue (Kobak & Berens, 2019), and the practice of analyzing latent variables has been both disputed (Chari & Pachter, 2023) and defended (Lause et al., 2024). This entire discussion could have been avoided if only distances between latent variables had been identifiable.

Providing identifiability guarantees has been heavily investigated. We survey key results in Secs. 2 and 6, but the executive summary is that current approaches require significant model restrictions (e.g. linearity assumptions), labeled data, or a combination of both. These solutions are underwhelm-

---

[1]Department of Applied Mathematics and Computer Science, Technical University of Denmark, Lyngby, Denmark [2]Department of Statistics, University of British Columbia. Correspondence to: Stas Syrota <stasy@dtu.dk>.

*Proceedings of the 42nd International Conference on Machine Learning*, Vancouver, Canada. PMLR 267, 2025. Copyright 2025 by the author(s).

ing to practitioners who often lack a labeling mechanism and are keen to leverage contemporary generative models.

**In this paper**, we change the identifiability question to arrive at working tools that do not require data labels and only impose minimal restrictions on the used models. Instead of identifying the latent variables, we identify the *relationship between latent variables*. For example, instead of identifying the coordinates of the latent variables we identify pairwise distances. In our experience, this matches the needs of domain experts who rarely assign meaning to the coordinates of latent variables. Using differential geometry, we prove strong identifiability guarantees on pairwise distances, angles, and more. We empirically validate our theory on four different generative models.

## 2. Background and notation

Before stating our main questions and results, we recap the prerequisite background information. We position our work relative to the existing literature in Sec. 6.

**Deep latent variable models** learn densities of data $X \in \mathcal{D}$ parametrized by latent variables $Z \in \mathcal{Z}$, such that $p(X) = \int p(X|Z)p(Z)\mathrm{d}Z$ (Tomczak, 2024). We consider models with *continuous* latent variables, i.e., $\mathcal{Z} \subseteq \mathbb{R}^n$. Examples of this model class include *probabilistic PCA* (Tipping & Bishop, 1999), *variational autoencoders* (Kingma & Welling, 2013; Rezende et al., 2014), *normalizing flows* (Tabak & Vanden-Eijnden, 2010; Lipman et al., 2022), *diffusion models* (Ho et al., 2020) and more.

Formally, we define a model as a tuple of random variables $(Z, X)$ where the latent $Z$ drives the observations $X$ through a measurable *generator function* $f : \mathcal{Z} \to \mathcal{D}$, often called *the decoder*, and a *noise mechanism* $h : \mathcal{D} \times \mathcal{D} \to \mathcal{D}$ that makes the relationship stochastic through a noise term $\epsilon$,

$$Z^i \sim P_Z, \quad \epsilon^i \sim P_\epsilon, \quad X^i = h(f(Z^i), \epsilon^i) \quad (1)$$

where $Z^i$ and $\epsilon^i$ are assumed independent. We further adopt a standard regularity assumption that $h$ and $P_\epsilon$ are such that $\epsilon^a \overset{\mathrm{d}}{=} \epsilon^b$, $h(f(Z^a), \epsilon^a) \overset{\mathrm{d}}{=} h(f(Z^b), \epsilon^b)$ if and only if $f(Z^a) \overset{\mathrm{d}}{=} f(Z^b)$. Here $\overset{\mathrm{d}}{=}$ denotes equality in distribution and the assumption ensures that the noise $\epsilon$ does not interfere with the causal relationship between $X$ and $Z$.

**Statistical model** arises when we learn the parameters of the generative model given realizations $\mathbf{x}$ of $X$. Learning the generative model means estimating its parameters $\theta = (f, P_Z)$, which represent the decoder and the latent distribution, respectively. These give rise to the marginal distribution of the data $P_\theta$ that quantifies model fit. Formally, we define a model $M$ as

$$M(\mathcal{F}, \mathcal{P}_Z) = \{P_\theta \text{ on } \mathcal{D} \mid \theta = (f, P_Z) \in \mathcal{F} \times \mathcal{P}_Z\}, \quad (2)$$

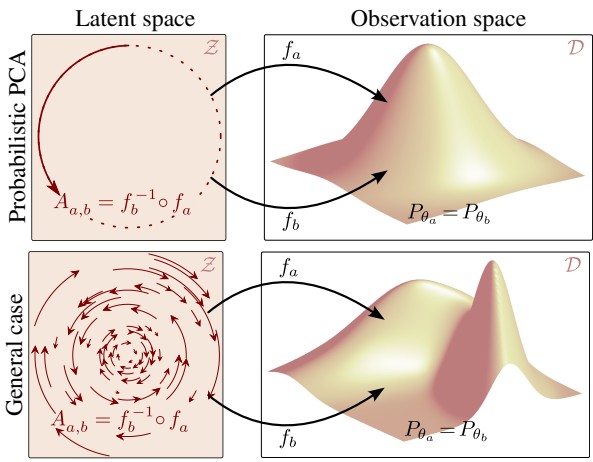

Figure 2: Indeterminacy transformations characterize the identifiability equivalence class. *Top row:* Probabilistic PCA has linear decoders, such that the indeterminacy transformations are rotations. *Bottom row:* In general deep latent variable models the indeterminacy transformations are the general class of diffeomorphisms acting on the latent space.

where $\mathcal{F}$ and $\mathcal{P}_Z$ are the sets of possible generator functions and distributions on the latent space, respectively. Designing a deep latent variable model means specifying $\mathcal{F}$ and $\mathcal{P}_Z$.

**Identifiability** concerns the uniqueness of parametrizations. We say that two parameters $\theta$ and $\theta'$ are equivalent, $\theta \sim \theta'$, if the resulting distributions $P_\theta$ and $P_{\theta'}$ are the same. The induced *equivalence class* is denoted $[\theta] = \{\theta' : P_\theta = P_{\theta'}\}$. Informally, this class captures the different ways in which a specific density can be parameterized. Following Xi & Bloem-Reddy (2023), we say that a model is *strongly identifiable* if $[\theta]$ is a singleton, i.e. the model parametrization is unique, while a model is *weakly identifiable* if it can be identified up to the equivalence class $[\theta]$.

As an example, in probabilistic PCA (Tipping & Bishop, 1999), the latent variables can only be identified up to an unknown rotation due to the rotational symmetry of the Gaussian distribution. We then write the equivalence class as $[\theta] = \{R\theta\}$, where $R$ is any rotation matrix (Fig. 2, top).

**Indeterminacy transformations** provide means to characterizing the equivalence class of a latent variable model $M$ (Xi & Bloem-Reddy, 2023). Given two parametrizations of a model $\theta_a = (f_a, P_{Z_a})$ and $\theta_b = (f_b, P_{Z_b})$ with resulting marginal distributions $P_{\theta_a} = P_{\theta_b}$, an *indeterminacy transformation* at $(\theta_a, \theta_b)$ is a measurable function $A_{a,b} : \mathcal{Z} \to \mathcal{Z}$ such that $P_{\theta_a} = P_{\theta_b}$ and $f_a \circ A_{a,b}^{-1} = f_b$; c.f. bottom panel of Fig. 2. Xi & Bloem-Reddy (2023) prove that the set of all indeterminacy transformations, denoted $\mathbf{A}(M)$, fully determines the equivalence class $[\theta]$. *This result establishes the equivalence between parameter identifiability and indeterminacy transformations of the latent*

*space and their associated decoders.*

**Identifiable task** captures latent computations with identifiable outcomes (Xi & Bloem-Reddy, 2023). Here, a task is defined by first *selecting* latent points $\mathbf{z}_n = s(\theta, \mathbf{x}_m) \in \mathcal{Z}$, and secondly by evaluating the task $t(\theta, \mathbf{x}_m, \mathbf{z}_n)$. The selection mechanism can e.g. be the inverse decoder, while a task could be independence testing in causal discovery or measuring the distance between latent representations.

Following Proposition 3.1 from Xi & Bloem-Reddy (2023), we can state the sufficient condition for the identifiability of a task in terms of indeterminacy transformations.

**Definition 2.1.** A task $(s, t)$ is identifiable up to $[\theta]$ if, for each $A \in \mathbf{A}(M)$ and $\mathbf{x}_m \in \mathcal{D}$ with $\mathbf{z}_n \in \mathcal{Z}$:

$$\begin{aligned} t(\theta, \mathbf{x}_m, \mathbf{z}_n) &= t(A\theta, \mathbf{x}_m, A(\mathbf{z}_n)) \\ \text{and } s(A\theta, \mathbf{x}_m) &= A(s(\theta, \mathbf{x}_m)), \end{aligned} \quad (3)$$

where with $\theta_a, \theta_b \in [\theta]$, we have $A\theta_a = \theta_b = (f_a \circ A^{-1}, A_\# P_{Z_a}) = (f_b, P_{Z_b})$ and $A_\# P_{Z_a}$ denotes the pushforward of the probability measure $P_{Z_a}$.

## 3. Problem statement

We address the challenge of making pairwise distances statistically identifiable in modern deep generative models without impractical assumptions. Next, we outline our assumptions and formalize the objective, which we solve in Sec. 4 and we show that our approach ensures not only identifiable distances, but also a broader set of identifiable metric structures.

### 3.1. Assumptions on $\mathcal{F}$ and $\mathcal{Z}$

Following typical literature (Xi & Bloem-Reddy, 2023; Shao et al., 2018), we further impose assumptions on the space of our decoder functions $\mathcal{F}$ and the latent space $\mathcal{Z}$. We consider decoders that are smooth functions $f : \mathcal{Z} \to \mathcal{D}$ such that for each $f \in \mathcal{F}$:

**A1** $\mathcal{Z}$ is compact

**A2** $f$ is injective

**A3** The differential of $f$, $\mathrm{d}f$, has full column rank

**A4** All $f \in \mathcal{F}$ have the same image. That is, for any $f_a, f_b \in \mathcal{F}$, we have $f_a(\mathcal{Z}) = f_b(\mathcal{Z}) := \mathcal{M} \subseteq \mathcal{D}$

Assumptions A2-A4 are repeated from Xi & Bloem-Reddy, whereas we add assumption A1 and require $f$ to be smooth. Together, these allow us to treat the image of the decoder as a smooth manifold. Assumption A1 is purely technical and can be interpreted as (after model training) we consider a compact subset of the latent space, e.g. the range of floating point numbers. Jointly, the assumptions may appear restrictive, but they are satisfied by contemporary models

such as $\mathcal{M}$-flows (Brehmer & Cranmer, 2020), normalizing flows, and diffusion models. VAEs need not satisfy A2. On the other hand, A3 can be empirically validated after model training (Shao et al., 2018), and experiments (Sec. 5) show that our methodology is effective in this setting.

### 3.2. Identifiability of distances

In this paper, we shift focus from identifiability of latent representations (or equivalently, model parameters) and instead identify the relations between them. As our main focus, we seek to establish a distance measure that is invariant under the indeterminacy transformations $\mathbf{A}(M)$ of a deep latent variable model and therefore identifiable.

**Problem 1.** Consider a deep latent variable model $M(\mathcal{F}, \mathcal{P}_Z)$ and $\mathbf{A}(M)$ its set of indeterminacy transformations. We want to identify latent distances, i.e. find a 'meaningful' distance function $d : \mathcal{Z} \times \mathcal{Z} \to \mathbb{R}_+$, such that given a parametrization $\theta$, for any $\mathbf{z}_1, \mathbf{z}_2 \in \mathcal{Z}$ and $A \in \mathbf{A}(M)$ the following is staisfied:

$$d(\mathbf{z}_1, \mathbf{z}_2) = d(A(\mathbf{z}_1), A(\mathbf{z}_2)) \quad (4)$$

The inclusion of 'meaningful' in the problem definition emphasizes that solutions can be constructed that satisfy Eq. 4 without being of particular value, e.g. the trivial metric

$$d(\mathbf{z}_1, \mathbf{z}_2) = \mathbb{I}(\mathbf{z}_1 \neq \mathbf{z}_2) \quad (5)$$

is identifiable, but reveals little about latent similarities. Instead, we want the distance to reflect and respect the underlying mechanisms behind the observed data.

## 4. Main results

**Strategy and results at a glance.** In the following, we show that distances, angles, volumes, and more, can be identified in latent variable models that satisfy the weak assumptions in the previous section. Our proof strategy is to connect *indeterminacy transformations* from the identifiability literature with *charts* from the differential geometry literature. Once this connection is in place, our results easily follow. Furthermore, we use the connection to show that identifying Euclidean distances in the latent space is either impossible or requires forcing the decoder to have zero curvature. Below we present results with proof sketches and leave details to Appendices A and B.

### 4.1. Identifiability via geometry

We begin by focusing on the family of decoders $\mathcal{F}$ and analyzing the properties of their image.

**Lemma 4.1.** *Let $\mathcal{Z}$ and $\mathcal{D}$ be two smooth manifolds and $f \in \mathcal{F}$, then $f$ is a smooth embedding and $f(\mathcal{Z}) \subset \mathcal{D}$ is a submanifold in $\mathcal{D}$. In particular, $f : \mathcal{Z} \to f(\mathcal{Z})$ is a diffeomorphism.*

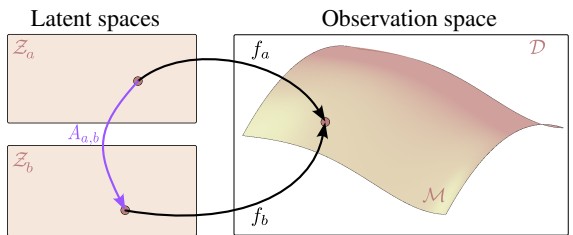

Figure 3: Decoders $f_a$ and $f_b$ parametrize the same manifold $\mathcal{M} \subset \mathcal{D}$ when $\theta_a = (f_a, P_{Z_a})$ and $\theta_b = (f_b, P_{Z_b})$ give the same marginal distributions $P_{\theta_a} = P_{\theta_b}$.

**Proof sketch.** Smoothness and assumption A3 lead to $f \in \mathcal{F}$ being a smooth map of constant rank (smooth immersion) while assumptions A1 and A2 make sure that the image of $f$ does not self-intersect. Given these properties, the claim follows from standard results in differential geometry.

One consequence of Lemma 4.1 is that given two trained models $\theta_a$ and $\theta_b$ with equivalent marginal distributions $P_{\theta_a} = P_{\theta_b}$, the resulting decoder functions $f_a, f_b$ act as reparametrizations of the same manifold $f_a(\mathcal{Z}) = f_b(\mathcal{Z}) = \mathcal{M}$. In particular the tuples $(f_a^{-1}, \mathcal{M})$ and $(f_b^{-1}, \mathcal{M})$ can be seen as coordinate charts of the manifold $\mathcal{M}$. This situation is the main subject of our analysis and is illustrated in Figure 3. In what follows, we will label the latent spaces by the associated models such that for $\theta_a$ we will have $\mathcal{Z}_a$ and similarly for $\theta_b$.

We will go between the different charts (or latent spaces) by using *generator transformations* in Definition 4.2 that push and pull along the respective decoders.

**Definition 4.2.** Given two equivalent parametrizations $\theta_a = (f_a, P_{Z_a})$ and $\theta_b = (f_b, P_{Z_b})$ of a model with $P_{\theta_a} = P_{\theta_b}$, we define the *generator transformation* $A_{a,b} : \mathcal{Z}_a \to \mathcal{Z}_b$ is

$$A_{a,b}(\mathbf{z}) = f_b^{-1} \circ f_a(\mathbf{z}), \qquad \text{for } \mathbf{z} \in \mathcal{Z}_a. \qquad (6)$$

Lemma 2.1 in Xi & Bloem-Reddy (2023) shows that any indeterminacy transformation $A \in \mathbf{A}(M)$ must be almost everywhere equal to the generator transformation. Whereas Xi & Bloem-Reddy focus on proving this result and using it for characterizing identifiability issues in general, we use this construction to show that it preserves the geometric properties of the manifold and, in particular, that the geodesic distance function (formally defined in Eq. 9) is invariant w.r.t. to the entire set $\mathbf{A}(M)$.

**Lemma 4.3.** *Given two equivalent parametrizations $\theta_a = (f_a, P_{Z_a})$ and $\theta_b = (f_b, P_{Z_b})$ of a model with $P_{\theta_a} = P_{\theta_b}$, the generator transformations $A_{a,b}(\mathbf{z})$ and $A_{a,b}^{-1}(\mathbf{z}) = A_{b,a}(\mathbf{z})$ are diffeomorphisms.*

**Proof sketch.** The result follows from the fact that a composition of diffeomorphisms is a diffeomorphism. The generator construction is only possible because of assumption A4.

From the perspective of differential geometry, if we are to collect all the coordinate charts of the form $(f^{-1}, \mathcal{M})$ stemming from the indeterminacy transformations in $\mathbf{A}(M)$ into a smooth atlas, then Lemma 4.3 tells us that the generator transforms play the of role of smooth transition maps between them, as illustrated in Fig. 3.

Lemma 4.1 tells us that the image of our decoder functions is a smooth manifold. Due to the 'Existence of Riemannian Metrics' result by Lee (2003), it admits a *Riemannian metric* $g$ that for each point $p$ on the manifold defines an inner product in its tangent space at $p$, denoted by $T_p\mathcal{M}$. The tuple $(\mathcal{M}, g)$ defines the Riemannian manifold structure that allows measurements on general smooth manifolds and is the theoretical foundation for our methodology.

For general data manifolds, there is neither a unique nor a known metric $g$. However, given a decoder function $f$ and a chosen metric (often Euclidean) $g^{\mathcal{D}}$ in the ambient space $\mathcal{D}$, we can construct the *pullback* metric $g^f$ in the latent space $\mathcal{Z}$ by pulling the ambient metric back to the latent space using the decoder.

**Definition 4.4.** Let $\mathcal{Z}$ be a smooth manifold and $(\mathcal{D}, g^{\mathcal{D}})$ be a Riemannian manifold. Furthermore, let $f : \mathcal{Z} \to \mathcal{M} \subseteq \mathcal{D}$ be a map satisfying assumptions A1-A3, the pullback metric $f^* g^{\mathcal{D}}$ on $\mathcal{Z}$ is defined as:

$$(f^* g^{\mathcal{D}})_p(u, v) = g^{\mathcal{D}}_{f(p)}(\mathrm{d}f_p(u), \mathrm{d}f_p(v)) \qquad (7)$$

for any tangent vectors $u, v \in T_p\mathcal{Z}$. In Eq. 7, $g^{\mathcal{D}}_{f(p)}$ means that we use ambient metric evaluated in the tangent space $T_{f(p)}\mathcal{M}$. The notation $\mathrm{d}f_p(u)$ means that the differential map of $f$ at $p \in \mathcal{Z}$ maps the vector $u \in T_p\mathcal{Z}$ to $f(u) \in T_{f(p)}\mathcal{M}$. We will denote the pullback metric as $g^f = f^* g^{\mathcal{D}}$ for shorter notation and let the domain of it be implicit from the definition of $f$.

The result of this important construction is that:

- it allows us to construct a Riemannian metric on $\mathcal{M}$ that respect the intrinsic properties of the Riemannian manifold $(\mathcal{M}, g)$. In this setting, the pullback metric $g^f$ represents some intrinsic $g$ in the coordinates defined by $\mathcal{Z}$ and $f$.

- we can make all the measurements from the latent space $\mathcal{Z}$ using $g^f$ as this construction makes the Riemannian manifolds $(\mathcal{Z}, g^f)$ and $(\mathcal{M}, g)$ the same, from a geometric perspective. Thus, we can concentrate our attention on $(\mathcal{Z}, g^f)$, while being consistent with $(\mathcal{M}, g)$ without worrying about $g$.

The pullback metric merely measures the length of a latent curve by first decoding the curve and measuring its length according to the data space metric. This is a quite 'meaningful' metric in line with the requirements of Problem 1.

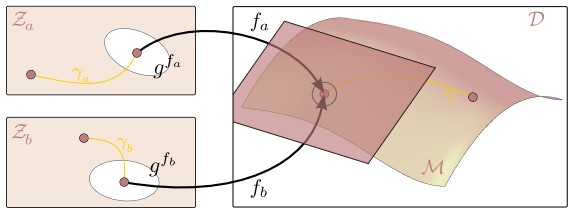

Figure 4: Pullback metrics assign a local inner product to latent spaces corresponding to measuring *along* the manifold spanned by the decoder. In the left panels, the white ellipsis corresponds to unit circles under the pullback metric, corresponding to a local Euclidean metric in the observation space $\mathcal{D}$. Geodesics (yellow curves) minimize length according to the pullback metric, corresponding to minimizing the length of the decoded curve along the manifold.

In the framework of pullback metrics defined by different decoders that span the same manifold, the generator transformations that comprise the space of indeterminacy transformations are isometries that preserve angles, length of curves, surface areas, and volumes on the manifold.

**Theorem 4.5.** *Let $\theta_a = (f_a, P_{Z_a})$ and $\theta_b = (f_b, P_{Z_b})$ with $P_{\theta_a} = P_{\theta_b}$ and let $(\mathcal{Z}_a, g^{f_a})$ and $(\mathcal{Z}_b, g^{f_b})$ be the associated Riemannian manifolds, then the generator transform is an isometry and it holds that:*

$$\left( A_{a,b} \right)^* g^{f_b} = g^{f_a} \qquad (8)$$

*Thus, making $(\mathcal{Z}_a, g^{f_a})$ and $(\mathcal{Z}_b, g^{f_b})$ isometric. This makes Riemannian geometric properties such as lengths of curves, angles, volumes, areas, Ricci curvature tensor, geodesics, parallel transport, and the exponential map identifiable.*

**Proof sketch.** First, we show that Eq. 8 is satisfied, establishing the isometry property, then we refer to results in Riemannian geometry to establish the isometric invariance of a particular property. To obtain identifiability, each property is expressed in terms of a task from Section 2 and Definition 2.1 is shown to be satisfied.

To solve Problem 1 we use the *geodesic distance* from Definition 4.6 and show that it is identifiable in Theorem 4.7.

**Definition 4.6.** Let $(\mathcal{Z}_a, g^{f_a})$ be a Riemannian manifold, then for $\mathbf{z}_1, \mathbf{z}_2 \in \mathcal{Z}$ we define the geodesic distance function

$$d_{g^{f_a}}(\mathbf{z}_1, \mathbf{z}_2) = \inf_\gamma \int_0^T |\gamma'(t)|_{g^{f_a}} \, dt \qquad (9)$$

where $\gamma : [0, T] \to \mathcal{Z}_a$ is a latent curve from $\mathbf{z}_1$ to $\mathbf{z}_2$.

Figure 4 illustrates how the geodesic distance measures the length of the shortest curve (geodesic) under the pullback metric. This is equivalent to finding the shortest curve *along* the manifold spanned by a decoder.

**Theorem 4.7.** *Let $\theta_a = (f_a, P_{Z_a})$ and $\theta_b = (f_b, P_{Z_b})$ with $P_{\theta_a} = P_{\theta_b}$ and let $A_{a,b}$ be the generator transform between the parameters. Furthermore, let $(\mathcal{Z}_a, g^{f_a})$ and $(\mathcal{Z}_b, g^{f_b})$ be the associated Riemannian manifolds. Then, the geodesic distance between $\mathbf{z}_1$ and $\mathbf{z}_2$ is identifiable and*

$$d_{g^{f_a}}(\mathbf{z}_1, \mathbf{z}_2) = d_{g^{f_b}}(A_{a,b}(\mathbf{z}_1), A_{a,b}(\mathbf{z}_2)) \qquad (10)$$

*for some $\mathbf{z}_1, \mathbf{z}_2 \in \mathcal{Z}_a$ be two points in the latent space that correspond to some $\mathbf{x}_1, \mathbf{x}_2 \in \mathcal{M}$ on the manifold.*

**Proof sketch.** First formulate the task of computing the geodesic distance in terms of Definition 2.1, then check the definition for the selection function by plugging in and for the task output by leveraging Theorem 4.5.

### 4.2. Identifiability of Euclidean distances

Theorem 4.7 represents our solution to Problem 1 and came as a result of treating the question of identifiability from the geometric perspective. While Section 6 outlines alternative approaches to the same problem, in the following we show that these must necessarily impose implicit constraint of flatness on the models.

**Proposition 4.8.** *Let $\mathcal{Z} = \mathbb{R}^n$ be the latent space and $(\mathcal{Z}, g^f)$ the associated Riemannian manifold. Furthermore, let $g^{E_p}$ denote a metric tensor that is proportional to the Euclidean metric tensor $g^E$, then:*

**P1** *If we choose $g^{E_p}$ as our metric in the latent space, that is equivalent to assuming $g^f = g^{E_p}$, then $(\mathcal{Z}, g^{E_p})$ can only be identifiable if the associated $f \in \mathcal{F}$ parametrize a flat manifold $\mathcal{M}$ within the ambient space $\mathcal{D}$, i.e. $\mathcal{M}$ has zero curvature.*

**P2** *If we choose the Euclidean distance to be identifiable, equivalent to assuming $g^f = g^E$, then P1 applies and the associated $f \in \mathcal{F}$ are such that the generator transforms are isometries of $\mathbb{R}^n$, i.e. translations, rotations, or reflections.*

*Proof.* Distance measures proportional to the Euclidean distance measure are characterized by the pullback metric $g^f$ being constant everywhere. If $g^f$ is constant everywhere, its directional derivatives vanish and the curvature is zero. The second point follows from Theorem 4.5 and standard linear algebra, e.g. Friedberg et al. (2014). $\qquad \square$

### 4.3. Main takeaways

Our results can be summarized by the following takeaways:

- The Riemannian metric space of a deep latent variable model is identifiable (Theorem 4.7) making distance measurements in the latent space identifiable. This solves Problem 1.

- Riemannian geometry properties of the learned manifold are identifiable (Theorem 4.5). Examples beyond distances include angles, volumes, and more. Jointly these provide a rich language for probabilistic data analysis in the latent space.

- Using Euclidean distances in the latent space is either not identifiable or must come at the cost of imposing flatness constraints on the model (Proposition 4.8).

- Any task whose identifiability boils down to the identifiability of the Riemannian metric is identifiable if the properties of the manifold allow.

**To exemplify the last point**, consider the Fréchet mean that generalizes the well-known mean to manifolds (Pennec, 2006). This is obtained by finding the point with minimal average squared distance to the data,

$$\mu_{\text{Fréchet}} = \operatorname*{argmin}_{\mathbf{z}_1 \in \mathcal{Z}} \sum_{i=1}^{N} d^2_{g^{fa}}(\mathbf{z}_1, \mathbf{z}_i) \qquad (11)$$

As the mean is defined as an optimization problem, there might exist multiple means, which violates the usual notion of identifiability. First, it is worth noting that the solution set is identifiable, although, in practice, one usually only computes a single optimum. In some situations, this singleton can, however, be identified based on properties of the manifold $\mathcal{M}$. Karcher (1977) and Kendall (1990) provide uniqueness conditions that connect the radius of the smallest geodesic ball containing the data with the maximal curvature of the manifold. Importantly, these are, principally, testable conditions such that it should be feasible to computationally test if a computed mean is identifiable. *Identifiability of some statistical quantities is, thus, within reach.*

## 5. Experiments

Identifiability is a theoretical concept studied in the asymptotic regime of infinite data. Our experiments aim to demonstrate that this asymptotic property is practically exploitable in standard models using off-the-shelf methods.

Theorem 4.7 proves that geodesic distances are identifiable, while Euclidean ones are not. This suggests that geodesic distances should be more stable under model retraining than the Euclidean counterpart. To test this hypothesis, we train 30 models with different initial seeds and compute both Euclidean ($d_E$) and geodesic ($d_g$) distances in the latent space between 100 randomly chosen unique point pairs ($\{p_i = (\mathbf{x}_j, \mathbf{x}_k)\}_{i=1}^{100}, \mathbf{x}_j, \mathbf{x}_k \in \mathcal{M}, j \neq k$) from the test set. We emphasize that the pairs are the same across all models, allowing us to measure the variances of the distances across models.

To assess the stability of the distance measures, we compute the coefficient of variation for each pair, which is evaluated

as $\mathrm{CV}(d_*(p_i))) = \sigma(d_*(p_i))/\mu(d_*(p_i))$ with $\sigma$ and $\mu$ denoting standard deviation and the mean across 30 seeds, and $d_*$ is a placeholder for the distance measure used ($d_E$ or $d_g$). The coefficient of variation is a unitless measure of variability, where low values indicate less variability. We use this to compare the variability of Euclidean and geodesic distances.

We consider two models and four datasets. First, a model that satisfies all our assumptions, and second, a model where we disregard the injectivity assumption. To compute geodesics we parametrize them by a spline connecting two points in the latent space and minimize its energy. The discussion around Definition A.23 in Appendix A covers how this leads to a geodesic. It has been noted that taking decoder uncertainty into account is key to good performance (Arvanitidis et al., 2018; Hauberg, 2018) and we follow the ensemble-based approach from Syrota et al. (2024), implying that we train an ensemble of 8 decoders. Details on computing geodesics and experimental details are in Appendices C and D. The code to reproduce our results is available in the project repository GitHub[1].

**MNIST and CIFAR10 with A1-A4 satisfied.** We use $\mathcal{M}$-flows (Brehmer & Cranmer, 2020) to construct a model with an injective decoder. We train this model on a 3-class subset of MNIST (Deng, 2012) with a 2D latent space for visualization purposes and full CIFAR10 (Krizhevsky et al.). An example of a geodesic curve from digit 7 to digit 0 from the test set is visualized in Fig. 5. The geodesic crosses class boundaries where they are well-explored by the model and offer little uncertainty.

The left side of Fig. 7 shows a histograms of the coefficient of variation for the 100 point-pairs, where we see a narrower distribution with both a lower mean and spread for geodesic distances. We perform a one-sided Student's $t$-test for the null hypothesis that geodesic distances vary less than the Euclidean (Table 1) and find strong evidence for the hypothesis. This demonstrates that identifiability improves reliability.

**FMNIST and CELEBA with A2 relaxed and A3 verified.** The general VAE model is known to be effective due to its flexible decoder parametrized by a neural network with arbitrary architecture that is not guaranteed to be globally injective (A2) nor to have full rank Jacobian (A3).

---

[1]https://github.com/mustass/identifiable-latent-metric-space

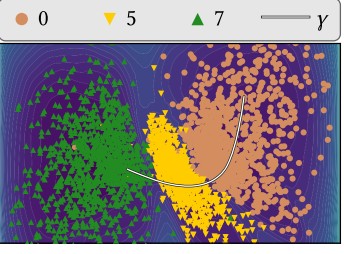

Figure 5: An example latent geodesic from a $\mathcal{M}$-flow model trained on three classes from MNIST. The background color indicates model uncertainty.

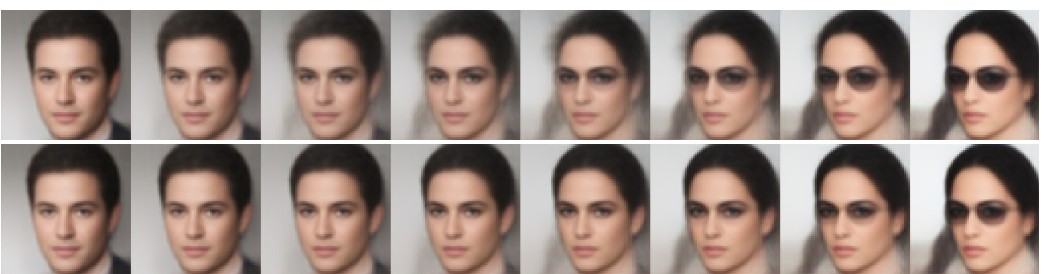

Figure 6: Geodesic (top) and Euclidean (bottom) interpolations, are highly similar, but distances still differ significantly (Fig. 7).

|            | MNIST | FMNIST | CIFAR10 | CELEBA |
|------------|-------|--------|---------|--------|
| $t$-statistic | -8.64 | -16.75 | -42.83 | -22.33 |
| $p$-value   | 1.00  | 1.00   | 1.00    | 1.00   |

Table 1: One-sided Student's $t$-test for the variability of geodesic versus Euclidean distances

We train this model on the FMNIST (Xiao et al., 2017) and CELEBA (Liu et al., 2015) datasets where the architecture is composed of convolutional and dense layers with ELU activation functions. The latent space dimension is 64 and we further employ Resize-Conv layers (Odena et al., 2016) to improve image quality. We follow the approach from Shao et al. (2018) to validate that the decoder Jacobian is, indeed, always full rank. An example geodesic is shown in Fig. 6 alongside a Euclidean counterpart, where we do not observe a significant difference between generated images.

Figs. 7b and 7d (right side) show that the coefficient of variation for geodesic distances has both lower mean and standard deviation than Euclidean distances. The one-sided Student's $t$-test again validates this observation (Table 1). This demonstrates that geodesic distances remain more reliable than Euclidean ones even when the injectivity assumption may be violated.

## 6. Relations to existing work

We connect questions of *identifiability* with results from *differential geometry*. To our knowledge, no previous studies have formally connected these otherwise disjoint fields. Our work is, however, linked to a large body of prior works.

**Identifiability** is well studied in the ICA or source separation literature (Comon, 1994; Hyvärinen & Pajunen, 1999). The analysis of identifiability in deep generative models stems from a connection between VAEs and ICA first noticed by Khemakhem et al. (2020). Many works either focus on formulating identifiability-enhancing constraints, typically placed on the decoder or latent distribution, or obtaining data from more diverse sources, e.g., multiple environments or multiple views (Kivva et al., 2022; Hyvärinen et al., 2019; Gresele et al., 2019; Locatello et al., 2020; 2019b; Shu et al., 2019). On the other hand, few works exist

that characterize when, and what types of non-identifiability are acceptable in deployments of deep generative models without significant constraints.

**Latent space geometries** have been studied in various contexts to define more 'meaningful' latent interpolations and distances (Tosi et al., 2014; Arvanitidis et al., 2018; Beik-Mohammadi et al., 2021). While Hauberg (2018) alludes to connections between latent space geometries and identifiability, no formal statement has previously been made. Our work, thus, brings further mathematical justification to the algorithmic tools developed for latent space geometries. Detlefsen et al. (2022) has previously demonstrated that latent space geometries recover evolutionary structures from models of proteins that are invisible under an Euclidean latent geometry. Our work adds credibility to these findings, which can now be seen as identifiable.

**Causality** strongly relies on identifiability as there is little point in recovering the 'true' causal model if it is not guaranteed to be unique. The goal of *causal representation learning* (Schölkopf et al., 2021) is closely linked to our quest for identifiable representations, or, at least, relationships between such. Our approach, however, is not immediately applicable to many questions of causality as these often amount to establishing *independence* between variables (Peters et al., 2017). Under the geometric lens, we do not have a canonical coordinate system in the latent space, which complicates splitting the latent space into factors to be considered independent.

**Transformations of the latent space** have proven valuable in various areas, including disentangled and equivariant learning, highlighting their mathematical and conceptual connections. Higgins et al. (2018); Wang et al. (2021); Zhu et al. (2021) assume a disentangled latent space where transformations decompose into individual factors of variation and the goal is to find representations that respect this structure. This can be interpreted as equivariance of the generator transformations $A_{a,b}$ (Definition 4.2). Instead of enforcing specific properties of $A_{a,b}$, our theory analyzes the natural properties of the generator transformations, and we find that just by learning a generative model, $A_{a,b}$, will automatically respect the latent Riemannian geometry (Theorem 4.5). Thus, our theory is valid regardless of whether a

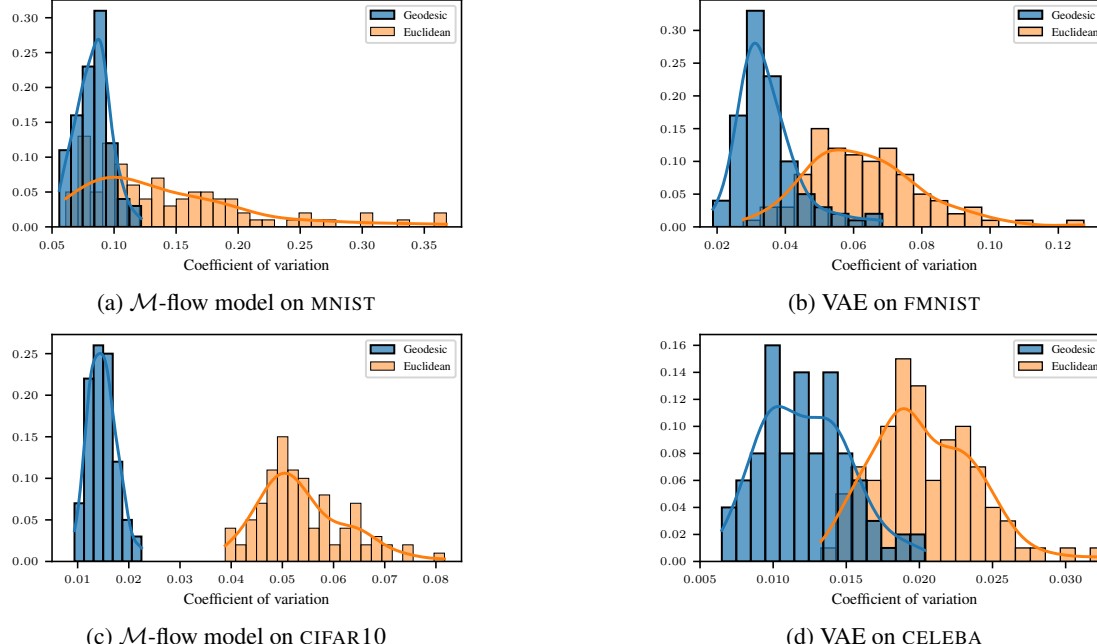

$(a)$ $\mathcal{M}$-flow model on MNIST

$(b)$ VAE on FMNIST

$(c)$ $\mathcal{M}$-flow model on CIFAR10

$(d)$ VAE on CELEBA

Figure 7: Histograms of coefficients of variation for Euclidean and geodesic distances on MNIST (7a), FMNIST (7b), CIFAR10 (7c) and CELEBA (7d). Geodesic distances vary significantly less, which is quantified in Table 1.

disentangled latent space exists in the sense of Higgins et al. (2018).

**Disentanglement** can be seen as a 'poor man's causality' (Detlefsen & Hauberg, 2019), where the key generating factors are sought to be axis-aligned in the latent space. This is generally known to be mathematically impossible (Locatello et al., 2019a), much in line with proposition 4.8. Similar to identifiability, this difficulty has been addressed by inductive biases (Bouchacourt et al., 2018) or (weak) data labels (Locatello et al., 2020; 2019b; Shu et al., 2019). Empirical studies, however, hint that some key factors can be recovered in practice (Higgins et al., 2016; Dittadi et al., 2020; Suter et al., 2019). Rolinek et al. (2019) noted that standard disentanglement pipelines only work when the variational distribution, parametrized by the encoder, is restricted to a diagonal covariance. This suggests that current results in disentanglement may be artifacts of a poor Bayesian approximation. Our work suggests that disentanglement is perhaps better achieved by looking for 'geometric factors', such as *principal geodesics* (Fletcher et al., 2004).

**Relative representations** are explicitly constructed from the original representations and a set of anchor points to be invariant under angle-preserving transformations (using cosine similarity) (Moschella et al., 2022) or isometries (using a proxy for geodesic distances) Yu et al. (2024). Our approach instead provides mathematical guarantees on invariance under isometries of the original representations and establishes the link to statistical identifiability.

## 7. Weaknesses and open questions

Our work provides strong identifiability guarantees for essential quantities such as pairwise distances in contemporary generative models. However, our approach is not problem-free and we highlight some pitfalls to be aware of.

**Observation metrics matter.** Our identifiable geometric structure relies on the idea of locally bringing the observation space metric into the latent space. This has many benefits but also raises the question of choosing the observation space metric. This choice will directly impact the final latent distances. Did we then replace one difficult problem (identifiability) with another (choosing observation metric)? We argue that most data is equipped with units of measurement, which greatly simplifies the task of picking a suitable metric in the observation space. Furthermore, being explicit about how data is compared improves the transparency of the conducted data analysis. Finally, we emphasize that we are *not* proposing to bring the Euclidean distance from observation space into the latent space, but only to do so *infinitesimally*, i.e. measuring *along* the manifold.

**Identifiability comes at a (computational) cost.** Euclidean distances are cheap to compute, unlike the geodesic distances we consider. For any geodesic distance, we must solve an iterative optimization problem. Fortunately, this is a locally convex problem, that only requires estimating a limited number of parameters, so the computation is feasible. Yet it remains significantly more costly than computing

a Euclidean distance. We argue that when identifiability is important, e.g. in scientific knowledge discovery and hypothesis generation, the additional computational resources are well-spent. Proposition 4.8 effectively tells us that we must choose between (cheap) flat decoders or (expensive) curved ones: we cannot get the best of both worlds.

**Compression matters.** Our current work rests on the assumption that the latent space has a dimension that is less than or equal to the data space dimension, i.e. $\dim(\mathcal{Z}) \leq \dim(\mathcal{D})$. This remains the standard setting for representation learning and generative models. However, if we, for the sake of argument, wanted to identify distances between the weights of overparametrized neural networks, then our strategies would not directly apply. Early work has begun to appear on understanding the geometric structure of overparametrized models (Roy et al., 2024), which suggests that perhaps our approach can be adapted.

**Injectivity remains an issue.** Most of our assumptions are purely technical and easily satisfied in practice. The key exception is Assumption A2 stating that the decoder $f$ must be *injective*. The decoders of contemporary models such as diffusion models and (continuous) normalizing flows (including those trained with flow matching) are injective and the assumption is satisfied. However, general neural networks cannot be expected to be injective, such that variational autoencoders and similar models are not identifiable 'out of the box'. We have demonstrated that (injective) $\mathcal{M}$-flow architectures can be used for such models, and empirically we observe that the geodesic distance increases robustness in non-injective models. This gives hope that theoretical statements can be made without the injectivity assumption. One such path forward may be to consider notions of *weak injectivity* (Kivva et al., 2022), which is less restrictive and more easily satisfied in practice.

## 8. Conclusion

In this paper, we show that latent distances, and similar quantities, can be statistically identified in a large class of generative models that includes contemporary models without imposing unrealistic assumptions. This is a significant improvement over existing work that tends to impose additional restrictions on either model or training data. Our results are significant when seeking to understand the mechanisms that drive the true data-generating process, e.g. in scientific discovery, where reliability is essential.

Practically, it is important to note that our strategy requires no changes to how models are trained. Our constructions are entirely *post hoc*, making them broadly applicable.

Our proof strategy relies on linking identifiability with Riemannian geometry; a link that does not appear to have formally been made elsewhere. This link paves a way forward

as many tools readily exist for statistical computations on manifolds. For example, Riemannian counterparts to *averages* (Karcher, 1977), *covariances* (Pennec, 2006), *principal components* (Fletcher et al., 2004), *Kalman filters* (Hauberg et al., 2013), and much more readily exist. In principle, it is also possible to devise computational tests to determine if these statistics are identifiable for a given model and dataset. This is, however, future work.

## Impact statement

This paper improves our collective understanding of which aspects of a statistical model can be identified. As the theoretical understanding translates directly into algorithmic tools, our work has an impact potential beyond the theoretical questions. Being able to identify relationships between latent representations of data can aid in the process of scientific discovery as we increase the reliability of the data analysis. Our work can also help provide robustness to interpretations of neural networks and other statistical models, which may help in explainability efforts.

## Acknowledgments

This work was supported by a research grant (42062) from VILLUM FONDEN. This project received funding from the European Research Council (ERC) under the European Union's Horizon research and innovation programme (grant agreement 101125993). The work was partly funded by the Novo Nordisk Foundation through the Center for Basic Machine Learning Research in Life Science (NNF20OC0062606). JX is supported by an Natural Sciences and Engineering Research Council of Canada (NSERC) Canada Graduate Scholarship. BBR acknowledges the support of NSERC: RGPIN2020-04995, RGPAS-2020-00095.

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

# A. Riemannian Geometry

This appendix covers the necessary concepts and results from differential geometry. For more full treatment, the reader is referred to excellent sources such as the book by Lee (2003).

The notation in this appendix is self-contained and separated from the main sections of the paper. Thus, the symbols that were used in the main body can be reused in the following but denote a different concept. The notation is introduced as we go along and should not lead to confusion.

To gradually build up the necessary constructions bottom-up, we start from the concept of a topology that we will morph into a Riemannian manifold by progressively adding structure.

## A.1. Topological concepts

A topology $\mathcal{T}$ on a set $X$ is a collection of subsets of $X$, that defines which sets are open in $X$. This gives rise to the notion of a neighborhood of a point $p \in X$ for an abstract set $X$.

**Definition A.1.** A topology $\mathcal{T}$ on $X$ is a collection of subsets of $\mathcal{P}(X)$ satisfying the following axioms:

1. $X$ and $\varnothing$ are in $\mathcal{T}$.

2. The union of any family of subsets in $\mathcal{T}$ are in $\mathcal{T}$.

3. The intersection of any finite family of subsets in $\mathcal{T}$ are in $\mathcal{T}$.

The tuple $(X, \mathcal{T})$ is called a topological space, and the elements of $\mathcal{T}$ are called open sets.

To construct a topology, we need a basis.

**Definition A.2.** A basis for a topology $\mathcal{T}$ on a set $X$ is a collection $\mathcal{B}$ of subsets of $X$ such that:

1. For each $x \in X$, there is at least one $B \in \mathcal{B}$ such that $x \in B$.

2. If $x \in B_1 \cap B_2$ for $B_1, B_2 \in \mathcal{B}$, then there is a $B_3 \in \mathcal{B}$ such that $x \in B_3 \subseteq B_1 \cap B_2$.

In cases where we work with subsets of $X$, we can use the topology of $X$ to define a topology on the subsets.

**Definition A.3** (Subspace Topology). Let $X$ be a topological space with topology $\mathcal{T}$ and $Y \subseteq X$. The subspace topology on $Y$ is defined as $\mathcal{T}_Y = \{Y \cap U | U \in \mathcal{T}\}$.

The notion of a topological leads to the construction of topological manifold that is the prerequisite to a Riemannian manifold.

**Definition A.4.** Suppose $M$ is a topological space. We say that $M$ is a topological manifold of dimension $n$ or a topological $n$-manifold if it has the following properties:

1. $M$ is a Hausdorff space: for every pair of distinct points $p, q \in M$, there are disjoint open subsets $U, V \subseteq M$ such that $p \in U$ and $q \in V$.

2. $M$ is second-countable: there exists a countable basis for the topology of $M$.

3. $M$ is locally Euclidean of dimension $n$: each point of $M$ has a neighborhood that is homeomorphic to an open subset of $\mathbb{R}^n$.

Definition A.4 specifies what we mean by a homeomorphism.

**Definition A.5.** Let $X$ and $Y$ be topological spaces, a map $F : X \to Y$ is

- continuous if the preimage of every open set in $Y$ is open in $X$.

$$\forall V \subseteq Y \text{ open } \Rightarrow F^{-1}(V) \subseteq X \text{ open}$$

- injective if $F(x) = F(y)$ implies $x = y$.

- surjective if for every $y \in Y$ there is an $x \in X$ such that $F(x) = y$.

- bijective if it is both injective and surjective.

- homeomorphism if it is bijective and both $F$ and $F^{-1}$ are continuous.

If $F$ is a homeomorphism, then $X$ and $Y$ are called homeomorphic. If $F$, on the other hand, is not bijective but only injective, we call $F$ a topological embedding.

**Definition A.6.** Let $X$ and $Y$ be topological spaces; a continuous injective map $F : X \rightarrow Y$ is called a topological embedding if it is a homeomorphism onto its image $F(X) \subseteq Y$ in the subspace topology.

The notion of local homeomorphism defined in the following is fundamental to the construction of a Riemannian manifold.

**Definition A.7.** For two topological spaces $X$ and $Y$, a continuous map $F : X \rightarrow Y$ is called a local homeomorphism if every point $p \in X$ has a neighborhood $U \subseteq X$ such that $F(U)$ is open in $Y$ and $F$ restricts to a homeomorphism from $U$ to $F(U)$.

Definition A.4 requires the existence of a local homeomorphism from $M$ to $\mathbb{R}^n$ for every point $p \in M$ defined on a neighborhood $U$ of $p$. Each such local homeomorphism with the corresponding restriction $U \in M$ is called a coordinate chart.

**Definition A.8.** A coordinate chart on $M$ is a pair $(U, \phi)$ where $U$ is an open subset of M and $\phi : U \rightarrow \hat{U}$ is a homeomorphism from $U$ to an open subset $\hat{U} = \phi(U) \subseteq \mathbb{R}^n$.

It follows from the definition of a topological manifold $M$ that every point $p \in M$ lies in the domain of some chart. $U$ is called coordinate domain and $\phi$ local coordinate map. The foundation of a smooth manifold is a maximal smooth atlas containing coordinate charts that are smoothly compatible.

**Definition A.9.** Let $M$ be a topological $n$-manifold. Two charts $(U, \phi)$ and $(V, \psi)$ are called smoothly compatible if $U \cap V = \varnothing$ or their transition map

$$\psi \circ \phi^{-1} : \phi(U \cap V) \rightarrow \psi(U \cap V)$$

is a diffeomorphism in case $U \cap V \neq \varnothing$.

In Definition A.9, note how the smoothness of the transition map can be analyzed in terms of the smoothness of maps between open subsets of $\mathbb{R}^n$, namely the domains of the associated coordinate charts.

**Definition A.10.** A diffeomorphism between two open subsets $U$ and $V$ of $\mathbb{R}^n$ is a bijective map $F : U \rightarrow V$ such that both $F$ and $F^{-1}$ are continuous and differentiable.

### A.2. Smooth manifolds

We are now in a position to define a smooth manifold.

**Definition A.11.** Let $M$ be a topological manifold,

- an atlas $\mathcal{A}$ is a collection of charts whose domains cover $M$

- a smooth atlas is an atlas $\mathcal{A}$ such that any two charts in $\mathcal{A}$ are smoothly compatible

- a maximal smooth atlas is a smooth atlas that is not properly contained in any larger smooth atlas

- $M$ together with a maximal smooth atlas $\mathcal{A}$ is called a smooth manifold denoted by $(M, \mathcal{A})$

In the following, we define smoothness for a map between two smooth manifolds.

**Definition A.12.** Let $M, N$ be smooth manifolds, and let $F : M \rightarrow N$ be any map. We say that $F$ is a smooth map if for every $p \in M$, there exist smooth charts $(U, \varphi)$ containing $p$ and $(V, \psi)$ containing $F(p)$ such that $F(U) \subseteq V$ and the composite map $\psi \circ F \circ \varphi^{-1}$ is smooth from $\varphi(U)$ to $\psi(V)$. This means that $\psi \circ F \circ \varphi^{-1}$ is a map between subsets of $\mathbb{R}^n$ and $\mathbb{R}^n$ and we can apply the usual real calculus.

Since a manifold does not have the usual operations of the Euclidean vector space, one way to construct a tangent vector space to a manifold $M$ at a point $p \in M$ is to define it in terms of a tangent vector to some curve $\gamma : I \rightarrow M$.

**Definition A.13.** Let $M$ be a smooth manifold, and let $\gamma_1, \gamma_2 : (-\epsilon, \epsilon) \to M$ be smooth curves in $M$. Suppose that $\gamma_1(0) = \gamma_2(0) = p \in M$, then $\gamma_1$ and $\gamma_2$ are said to be equivalent if the following holds:

$$\frac{d(\varphi \circ \gamma_1)}{dt}\bigg|_{t=0} = \frac{d(\varphi \circ \gamma_2)}{dt}\bigg|_{t=0}$$

This defines an equivalence relation on the set of all smooth curves through $p$, and the equivalence classes are called tangent vectors of $M$ at $p$. The tangent space $T_pM$ to $M$ at $p$ is then defined as the set of all tangent vectors at $p$ and does not depend on the choice of the coordinate chart $\varphi$.

A vector space structure on the tangent space $T_pM$ is defined by using the coordinate charts that map between subsets of $\mathbb{R}^n$ and allow us to do vector addition and scalar multiplication. Lee (2003) shows that the resulting construction is independent of the choice of the charts and that $T_pM$ is an $n$-dimensional real vector space.

Considering maps between manifolds, we want to link the tangent space of one with the tangent space of the other. This is done by defining the differential $\mathrm{d}F$ of $F$ at a point $p$, which is a linear mapping from one manifold's tangent space to another's.

**Definition A.14.** Let $M$ and $N$ be smooth manifolds and $F : M \to N$ be a smooth map. For each $p \in M$ we define a map:

$$\mathrm{d}F_p : T_pM \to T_{F(p)}N$$

called the differential of $F$ at $p$, which is a linear map between the tangent spaces. The following property defines the differential:

$$\mathrm{d}F_p(v) = \frac{d(\varphi \circ \gamma)}{dt}\bigg|_{t=0}$$

where $\gamma$ is smooth a curve in $M$ through $p$ with $\gamma'(0) = v$ and $\varphi$ is a coordinate chart around $p$. This construction is independent of the choice of the chart $\varphi$ as shown in (Lee, 2003).

The differential allows us to assess the rank of a map between two manifolds.

**Definition A.15.** Given two smooth manifolds $M$ and $N$ a map $F : M \to N$ has constant rank $r$ at $p \in M$ if the linear map $dF_p : T_pM \to T_{F(p)}N$ has rank $r$. $F$ is called a smooth submersion if its differential is surjective at each point (rank $F = \dim N$). It is called a smooth immersion if its differential is injective at each point (rank $F = \dim M$)

To define new submanifolds as images of maps, we need the concept of a smooth embedding.

**Definition A.16.** Let $M$ and $N$ be smooth manifolds, a smooth embedding of $M$ into $N$ is a smooth immersion $F : M \to N$ that is also a topological embedding, i.e., a homeomorphism onto its image $F(M) \subseteq N$ in the subspace topology.

Theorem A.17 tells us when an injective smooth immersion is also a smooth embedding.

**Theorem A.17.** *(Proposition 4.22 in (Lee, 2003)) Let $M$ and $N$ be smooth manifolds, and $F : M \to N$ is an injective smooth immersion. If any of the following holds, then $F$ is a smooth embedding.*

- *$F$ is an open or closed map.*

- *$F$ is a proper map.*

- *$M$ is compact.*

- *$M$ has empty boundary and $\dim M = \dim N$*

Theorem A.18 tells us that images of smooth embeddings are submanifolds with smooth properties.

**Theorem A.18.** *(Proposition 5.2 in (Lee, 2003)) Suppose $M$ and $N$ are smooth manifolds and $F : M \to N$ is a smooth embedding. Let $S = F(M)$. With the subspace topology, $S$ is a topological manifold, and it has a unique smooth structure, making it into an embedded submanifold of $N$ with the property that $F$ is a diffeomorphism onto its image.*

## A.3. Riemannian manifolds

Proposition 13.3 of (Lee, 2003) proves the existence of a Riemannian metric $g$ in any smooth manifold $N$, where $g$ is a smooth, symmetric covariant 2-tensor field on $M$ that is positive definite at each point $p \in N$ and defines an inner product in a tangent space $T_p N$. The tuple $(N, g)$ is called a Riemannian manifold.

Often, in modeling situations, we think of Theorem A.18, that is, we imagine an embedded submanifold $S \subseteq N$ in some ambient space $N$. In this case, we say that $F : M \to N$ is a parametrization of $S$. If there is a Riemannian metric $g^N$ on $N$, there is a way to measure lengths of vectors in a tangent space to a point on $S$ using $g^N$ as it is embedded in this larger vector space that has a metric. We can use this to construct a metric on $S$ by pulling $g^N$ by $F$.

**Definition A.19.** For a map $F : M \to F(M) = S \subset N$ between manifolds $M$ and $S$, and a metric $g^N$ on $N$, the pullback metric $f^* g^N$ on $M$ is defined as:

$$(F^* g^N)_p(u, v) = g^N_{F(p)}(\mathrm{d}F_p(u), \mathrm{d}F_p(v)) \tag{12}$$

for any tangent vectors $u, v \in T_p M$.

Given two Riemannian manifolds, we can check if they are isometric by using the pullback metric.

**Definition A.20.** Given Riemannian manifolds $(M, g^M)$ and $(N, g^N)$, a smooth map $F : M \to N$ is called an isometry if $F$ is a diffeomorphism such that:

$$F^* g^N = g^M \tag{13}$$

In which case we say $(M, g^M)$ and $(N, g^N)$ are isometric

A series of results in (Lee, 2003) Chapters 13,15,16 show that if two manifolds are isometric through a diffeomorphism $F$, then $F$ preserves lengths of curves, distances, angles, volumes, and other geometric properties between manifolds.

**Definition A.21.** Given a Riemannian manifold $(M, g)$ we can define the following:

- length or norm of a tangent vector $v \in T_p M$ is defined to be

$$|v|_g = \langle v, v \rangle_g^{1/2} = g_p(v, v)^{1/2}.$$

- angle between two nonzero tangent vectors $v, w \in T_p M$ is the unique $\theta \in [0, \pi]$ satisfying

$$\cos \theta = \frac{\langle v, w \rangle_g}{|v|_g |w|_g}.$$

- tangent vectors $v, w \in T_p M$ are said to be orthogonal if $\langle v, w \rangle_g = 0$. This means either one or both vectors are zero, or the angle between them is $\pi/2$.

- given a smooth curve $\gamma : [a, b] \to M$ we can define the length of $\gamma$ to be:

$$\mathrm{L}_g(\gamma) = \int_a^b |\gamma'(t)|_g \, dt$$

and the energy of $\gamma$ to be:

$$\mathrm{E}_g(\gamma) = \frac{1}{2} \int_a^b |\gamma'(t)|_g^2 \, dt$$

Proposition 13.25 of Lee (2003) shows that given a curve $\gamma : [a, b] \to M$ and a reparametrization $u : [c, d] \to [a, b]$ that is a diffeomorphism, the length of the curve $\tilde{\gamma} = \gamma \circ u$, $\mathrm{L}_g(\tilde{\gamma})$ is equal to $\mathrm{L}_g(\gamma)$.

The notion of the length of a curve given in Definition A.21 allows us to consider the Riemannian distance from $p$ to $q$ $(p, q \in M)$ denoted by $d_g(p, q)$ and defined to be the infimum of $\mathrm{L}_g$ over all piecewise smooth curve segments from $p$ to $q$. A shortest curve is not unique since $\mathrm{L}_g(\gamma)$ is reparametrization invariant, and a set of curves is locally minimizing $\mathrm{L}_g(\gamma)$. One useful parametrization is by arc-length $s$, which ensures that we move along the curve at a constant speed.

**Definition A.22** (Arc-length parametrization). A curve $\gamma : [a, b] \to M$ is said to be parametrized by arc-length if the length of the curve between any two points $t_1$ and $t_2$ is equal to the difference in the parameter values $t_2 - t_1$. Formally, $\gamma$ is parametrized by arc-length if:

$$|\gamma'(t)|_g = 1$$

for all $t \in [a, b]$.

Curves $\gamma$ that are locally minimizing $\mathrm{L}_g(\gamma)$ and are parametrized by arc-length are called geodesics defined in Definition A.23.

**Definition A.23.** Given a Riemannian manifold $(M, g)$ and $x, y \in M$ with $x \neq y$, a geodesic curve on between $x$ and $y$ on $M$ is formally defined as a curve $\gamma : I \to M$ that locally minimizes the energy functional

$$\mathrm{E}_g(\gamma) = \frac{1}{2} \int_a^b |\gamma'(t)|_g^2 \, dt$$

over all smooth curves $\gamma : [a, b] \to M$ connecting two given points

$\gamma(a) = x$ and $\gamma(b) = y$, where $g$ is the Riemannian metric tensor on $M$.

As discussed by, e.g., Hauberg (2024) it is a standard result that a minimizer of the energy functional will necessarily be arc-length parametrized and minimize the length functional.

We can consider the Fréchet mean and variance of a set of points on a manifold.

**Definition A.24.** Let $(M, d_g)$ be a Riemannian metric space and let $\{x_1 \ldots x_N\} \in M$ be points on the manifold. For any point $p \in M$, Fréchet variance is defined to be:

$$\Psi(p) = \sum_{i=1}^N d_g^2(p, x_i)$$

Karcher means are the points $m \in M$ that locally minimize $\Psi$:

$$m = \arg\min_{p \in M} \sum_{i=1}^N d_g^2(p, x_i)$$

If there exists a unique $m \in M$ that globally minimizes $\Psi$, then it is a Fréchet mean.

## B. Proofs

**Theorem B.1.** *(Lemma 4.1 in the main text)*

*Let $\mathcal{Z}$ and $\mathcal{D}$ be two smooth manifolds and $f \in \mathcal{F}$, then $f$ is a smooth embedding and $f(\mathcal{Z}) \subset \mathcal{D}$ is a submanifold in $\mathcal{D}$. In particular, $f : \mathcal{Z} \to f(\mathcal{Z})$ is a diffeomorphism.*

*Proof.* By Definition A.15 $f$ is a smooth immersion as it is a smooth map of constant rank with injective differential (assumptions A2-A3). Using Theorem A.17 with the fact that $\mathcal{Z}$ is a compact set (assumption A1) gives us that $f$ is a smooth embedding. Finally, Theorem A.18 gives us that $f(\mathcal{Z})$ is a submanifold of $\mathcal{D}$ and $f$ is a diffeomorphism on its image. □

**Theorem B.2.** *(Lemma 4.3 in the main text) Let $f_a, f_b \in \mathcal{F}$ and consider the generator transform $A_{a,b} : \mathcal{Z}_a \to \mathcal{Z}_b$ defined by*

$$A_{a,b}(z) = f_b^{-1} \circ f_a(z)$$

*Then $A_{a,b}(z)$ and $A_{a,b}^{-1}(z) = A_{b,a}(z) = f_a^{-1} \circ f_b(z)$ are diffeomorphisms.*

*Proof.* The result follows from Theorem B.1 with the fact that $f_a, f_b \in \mathcal{F}$ have the same image due to assumption A4. □

**Theorem B.3.** *(Theorem 4.5 in the main text)*

*Let $\theta_a = (f_a, P_{Z_a})$ and $\theta_b = (f_b, P_{Z_b})$ with $P_{\theta_a} = P_{\theta_b}$ and let $(\mathcal{Z}_a, g^{f_a})$ and $(\mathcal{Z}_b, g^{f_b})$ be the associated Riemannian manifolds, then the generator transform is an isometry and it holds that:*

$$(A_{a,b})^* g^{f_b} = g^{f_a} \tag{14}$$

*Thus, making $(\mathcal{Z}_a, g^{f_a})$ and $(\mathcal{Z}_b, g^{f_b})$ isometric. This makes Riemannian geometric properties such as lengths of curves, angles, volumes, areas, Ricci curvature tensor, geodesics, parallel transport, and the exponential map identifiable.*

*Proof.* We first show that the generator transform is an isometry. Let us recall that by definition of the pullback we have:

$$g_p^{f_b}(u, v) = g_{f_b(p)}^{\mathcal{D}}(\mathrm{d}f_{b,p}(u), \mathrm{d}f_{b,p}(v))$$

for $u, v \in T_p\mathcal{Z}_b$ and where $\mathrm{d}f_{b,p}(v)$ denotes the differential of the map $f_b$ at a point $p \in \mathcal{Z}_b$ evaluated on the vector $v \in T_p\mathcal{Z}_b$. Using this, we will check Eq. 14 directly:

$$
\begin{aligned}
\left((A_{a,b})^* g^{f_b}\right)_p (y, w) &= g_{f_b \circ f_b^{-1} \circ f_a(p)}^{\mathcal{D}} \left(\mathrm{d}f_{b, f_b^{-1} \circ f_a(p)}(y), \mathrm{d}f_{b, f_b^{-1} \circ f_a(p)}(w)\right) \\
&= g_{f_a(p)}^{\mathcal{D}} \left(\mathrm{d}(f_b \circ f_b^{-1} \circ f_a)_p(y), \mathrm{d}(f_b \circ f_b^{-1} \circ f_a)_p(w)\right) \\
&= g_{f_a(p)}^{\mathcal{D}} \left(\mathrm{d}f_{a,p}(y), \mathrm{d}f_{a,p}(w)\right) \\
&= g^{f_a}
\end{aligned}
\tag{15}
$$

for $y, w \in T_p\mathcal{Z}_a$.

Since we have shown that the generator transform is an isometry, we can conclude that $(\mathcal{Z}_a, g^{f_a})$ and $(\mathcal{Z}_b, g^{f_b})$ are isometric and thus their Riemannian metric properties are identical (O'Neill, 1997)[Chapters 6 and 7]. To connect to identifiability, we express any of the properties as a task of the form described in Section 2 and use the isometry result above to conclude that the output will be same. We will show an example of this and prove the claim for the goedesic distance function (Theorem 4.7 in the main text) below. □

**Theorem B.4.** *(Theorem 4.7 in the main text)*

*Let $\theta_a = (f_a, P_{Z_a})$ and $\theta_b = (f_b, P_{Z_b})$ with $P_{\theta_a} = P_{\theta_b}$ and let $A_{a,b}$ be the generator transform between the parameters. Furthermore, let $(\mathcal{Z}_a, g^{f_a})$ and $(\mathcal{Z}_b, g^{f_b})$ be the associated connected Riemannian manifolds. Then, the geodesic distance between $\mathbf{z}_1$ and $\mathbf{z}_2$ is identifiable and it holds that:*

$$d_{g^{f_a}}(\mathbf{z}_1, \mathbf{z}_2) = d_{g^{f_b}}(A_{a,b}(\mathbf{z}_1), A_{a,b}(\mathbf{z}_2)) \tag{16}$$

*for some $\mathbf{z}_1, \mathbf{z}_2 \in \mathcal{Z}_a$ be two points in the latent space that correspond to some $\mathbf{x}_1, \mathbf{x}_2 \in \mathcal{M}$ on the manifold.*

*Proof.* We need to check Definition 2.1 to show that the task of measuring distances in the latent space is identifiable using the geodesic distance function.

Let $\mathbf{x}_1, \mathbf{x}_2 \in \mathcal{D}$ be the observed data points and consider the inverse of the decoders as our selection function given $\mathbf{x}_1$:

$$\mathbf{z}_1^a = s(\theta_a, \mathbf{x}_1) = f_a^{-1}(\mathbf{x}_1) \text{ and } s(A\theta, \mathbf{x}_1) = s(\theta_b, \mathbf{x}_1) = f_b^{-1}(\mathbf{x}_1) = \mathbf{z}_1^b \tag{17}$$

in a similar way we obtain $\mathbf{z}_2^a$ and $\mathbf{z}_2^b$. Define the task of measuring distances on the manifold as:

$$t(\theta_a, \{\mathbf{x}_1, \mathbf{x}_2\}, \{\mathbf{z}_1^a, \mathbf{z}_2^a\}) = d_{g^{f_a}}(\mathbf{z}_1^a, \mathbf{z}_2^a)$$
$$t(\theta_b, \{\mathbf{x}_1, \mathbf{x}_2\}, \{\mathbf{z}_1^b, \mathbf{z}_2^b\}) = d_{g^{f_b}}(\mathbf{z}_1^b, \mathbf{z}_2^b) \tag{18}$$

To check the selection function, take with $A \in \mathbf{A}(M)$ and some $\mathbf{x}_1 \in \mathcal{D}$. Then,

$$\begin{aligned}
\mathbf{z}_1^b = s(\theta_b, \mathbf{x}_1) &= f_b^{-1}(\mathbf{x}_1) \\
&= f_b^{-1} \circ f_a \circ f_a^{-1}(\mathbf{x}_1) \\
&= A_{a,b}(s(\theta_a, \mathbf{x}_1)) \\
&= A(s(\theta_a, \mathbf{x}_1)) \\
&= A_{a,b}(f_a^{-1}(\mathbf{x}_1)) \\
&= A_{a,b}(\mathbf{z}_1^a)
\end{aligned} \tag{19}$$

where we have used that $A$ is almost everywhere equal to the generator transform $A_{a,b}$ due to Xi & Bloem-Reddy.

To check the task function, let us recall that:

$$d_{g^{f_a}}(\mathbf{z}_1^a, \mathbf{z}_2^a) = \inf_\gamma \int_a^b |\gamma'(t)|_{g^{f_a}} dt \tag{20}$$

where $\gamma : [c, d] \to \mathcal{Z}_a$ is a curve in $\mathcal{Z}_a$ connecting $\mathbf{z}_1^a$ and $\mathbf{z}_2^a$ such that $\gamma(c) = \mathbf{z}_1^a$ and $\gamma(d) = \mathbf{z}_2^a$.

As a geodesic is not unique, multiple curves result in the infimum in Eq. 20. Let $\gamma_a$ be one solution, then it is by Definition A.23 a geodesic curve. By Theorem 4.5 we have that $(\mathcal{Z}_a, g^{f_a})$ and $(\mathcal{Z}_b, g^{f_b})$ are isometric, which means that $A_{a,b}$ maps geodesics to geodesics (O'Neill, 1997), then $\gamma_b : [\tilde{c}, \tilde{d}] \to \mathcal{Z}_b$ constructed from $\gamma_a$ by $A_{a,b}(\gamma_a)$ is a geodesic in $\mathcal{Z}_b$ and thus a solution for $d_{g^{f_b}}(A_{a,b}(\mathbf{z}_1^a), A_{a,b}(\mathbf{z}_2^a)) = d_{g^{f_b}}(\mathbf{z}_1^b, \mathbf{z}_2^b)$. $\square$

## C. Computing geodesics

A geodesic between $a$ and $b$ is defined to be a curve $\gamma(t)$ defined on some interval (usually $[0, 1]$) such that $\gamma(0) = a$ and $\gamma(1) = b$ minimizing the length functional defined in Definition A.21. In our work, we choose to parametrize a geodesic by a cubic spline (Schoenberg, 1946) and optimize the energy functional defined in Definition A.21 with respect to the free parameters using gradient methods.

### C.1. Gedoesic parametrized by a cubic spline

Having settled on a cubic spline as a parametrization of a geodesic, we will now describe the construction of the spline and use it to derive the free parameters of the resulting curve that we can use when minimizing the energy of that curve.

A cubic spline is a piecewise function with pieces that are cubic polynomials. The points where the pieces meet are called the knots, $h$, and we want to construct a continuous spline with continuous first and second derivatives. Individual components are polynomials, so we only need to constrain their behavior at the knots to satisfy the requirements. Suppose the knots are known, and we are using the splines to interpolate a set of points. In that case, these constraints and boundary constraints will usually give a system of linear equations that can be solved to find the coefficients of the polynomials. In our setting, however, we are using splines to define a path between two points, and the knots are the unknown parameters of the problem, as well as the coefficients of the polynomials. Furthermore, given that $a, b \in \mathbb{R}^n$, we will look to parametrize a geodesic curve $\gamma(t) = (\gamma_1(t), \ldots, \gamma_n(t)) \in \mathbb{R}^n$ and thus we will have $n$ splines, one for each dimension. In the following, we will describe how such a construction works for one dimension and invite the reader to conceptually repeat this for $n$ dimensions.

Following the idea of (Hauberg, 2024), we will start by connecting two points in the latent space ($a, b \in \mathbb{R}$) by a straight line $l : [0, 1] \to \mathbb{R}$ defined as $l(t) = a + t(b - a)$ and then find a cubic spline that will start and end in 0 to parametrize a deviation from the line. The result will be a curve $\gamma(t) = l(t) + S(t)$ that will connect the two points on the manifold.

The spline $S(t)$ is defined as a piecewise function with $n$ cubic polynomials with coefficients $a_i, b_i, c_i, d_i \in \mathbb{R}$, each defined on an interval $[h_i, h_{i+1}]$ where $h_i$ are the knots with $h_0 = 0$ and $h_n = 1$ set to be the endpoints.

$$
S(t) = \begin{cases}
S_1(t) & \text{if } t \in [h_0, h_1] \\
S_2(t) & \text{if } t \in [h_1, h_2] \\
\vdots & \vdots \\
S_n(t) & \text{if } t \in [h_{n-1}, h_n]
\end{cases}
\tag{21}
$$

where each $S_i(t)$ is a cubic polynomial:

$$
S_i(t) = a_i + b_i(t) + c_i(t)^2 + d_i(t)^3 \quad \text{for } t \in [h_{i-1}, h_i]
\tag{22}
$$

In the following, let $\xi = (a_1, b_1, c_1, d_1, \ldots, a_n, b_n, c_n, d_n)$ be a vector of all coefficients of the polynomials in our spline and $\xi[i, j]$ be a subvector of $\xi$ containing the coefficients of the $i$-th and $j$-th polynomial.

Boundary conditions mean that we need our first polynomial to start in $(0, 0)$ and the last polynomial to end in $(1, 0)$. This gives us two equations:

$$
S_1(0) = a_1 = 0 \quad \text{and} \quad S_n(1) = a_n + b_n + d_n + c_n = 0
\tag{23}
$$

which we translate into the following matrix equation of the coefficients $\xi$ and a $2 \times 4n$ matrix $B$:

$$
B\xi^T = \begin{bmatrix} 0 \\ 0 \end{bmatrix}
\tag{24}
$$

where

$$
B = \begin{bmatrix}
1 & 0 & 0 & 0 & \ldots & 0 & 0 & 0 & 0 & 0 \\
0 & 0 & 0 & 0 & \ldots & 0 & 1 & 1 & 1 & 1
\end{bmatrix}
\tag{25}
$$

The continuity conditions are met when the values at the knots are the same for the two meeting polynomials. This can be expressed as:

$$
S_i(h_i) = S_{i+1}(h_i) \Leftrightarrow S_i(h_i) - S_{i+1}(h_i) = 0 \quad \text{for } i = 1, \ldots, n - 1
\tag{26}
$$

and for each knot we can write this as a dot product of the coefficients $\xi[i, i+1]$ and a vector $c_i^0$:

$$
c_i^0 = \begin{bmatrix} 1 & h_i & h_i^2 & h_i^3 & -1 & -h_i & -h_i^2 & -h_i^3 \end{bmatrix}
\tag{27}
$$

such that the condition at a knot $i$ becomes:

$$
c_i^0 \xi[i, i+1]^T = 0
\tag{28}
$$

The conditions of first and second derivatives being continuous can be expressed in a similar way.

$$
\begin{aligned}
S_i'(h_i) = S_{i+1}'(h_i) &\Leftrightarrow S_i'(h_i) - S_{i+1}'(h_i) = 0 \quad \text{for } i = 1, \ldots, n - 1 \\
S_i''(h_i) = S_{i+1}''(h_i) &\Leftrightarrow S_i''(h_i) - S_{i+1}''(h_i) = 0 \quad \text{for } i = 1, \ldots, n - 1
\end{aligned}
\tag{29}
$$

and we can write these conditions as dot products of the coefficients $\xi[i, i+1]$ and vectors $c_i^1$ and $c_i^2$:

$$
c_i^1 = \begin{bmatrix} 0 & 1 & 2h_i & 3h_i^2 & 0 & -1 & -2h_i & -3h_i^2 \end{bmatrix}
\tag{30}
$$

$$
c_i^2 = \begin{bmatrix} 0 & 0 & 2 & 6h_i & 0 & 0 & -2 & -6h_i \end{bmatrix}
\tag{31}
$$

such that the conditions at a knot $i$ become:

$$
\begin{aligned}
c_i^1 \xi[i, i+1]^T = 0 \\
c_i^2 \xi[i, i+1]^T = 0
\end{aligned}
\tag{32}
$$

Having defined the smoothness constraints for a given knot $i$ we can construct matrices $C^0, C^1, C^2$ each with dimensions $(n-1) \times 4n$ where each row $i$ corresponds to the respective constraint at the knot $i$ with $4 \cdot (i-1)$ zeros before the constraint and $4 \cdot (n-1-i)$ zeros after the constraint. E.g. for $n = 4$ the $C^0$ matrix would look as follows:

$$C^0 = \begin{bmatrix} c_1^0 & 0 & 0 & 0 & 0 & 0 & 0 & 0 & 0 \\ 0 & 0 & 0 & 0 & c_2^0 & 0 & 0 & 0 & 0 \\ 0 & 0 & 0 & 0 & 0 & 0 & 0 & 0 & c_3^0 \end{bmatrix}$$

where each $c_i^0$ is a row vector as defined in Eq. 27.

Now, the final system of equations can be written as:

$$\underbrace{\begin{bmatrix} B \\ C^0 \\ C^1 \\ C^2 \end{bmatrix}}_{:=A} \xi^T = \mathbf{0} \tag{33}$$

resulting in $4n - 2$ equations for $4n$ unknowns. To solve this system of equations in an interpolation setting, we usually impose two additional constraints to get a square system of equations. These constraints can be that the second derivative is zero at the endpoints or that the second derivatives at the first and last knots are equal. The former is known in the literature as a natural spline and the latter as not-a-knot spline (Kress, 2012).

Getting back to our original task of finding free parameters of the curve that we can optimize its energy with respect to, we note that given that we have $4n$ coefficients, the actual number of free parameters is considerably smaller due to the constraints. The problem in Eq. 33 is known as the problem of finding the Null Space of the row space of matrix $A$. A basis for such null space, denoted by $\mathcal{N}(A)$, can be found by computing the Singular Value Decomposition (SVD) (Klema & Laub, 1980) of $A$. If $A$ is of rank $r$, then SVD of A is given by $A = U\Sigma V^T$ where $U$ and $V$ are orthogonal matrices with dimensions $((4n-2) \times 4n)$ each and $\Sigma$ is a $(4n \times 4n)$ diagonal matrix with $r$ nonzero singular values in the diagonal. The null space of $A$ is then given by the columns of $V^T$ corresponding to the zero singular values. Treating $\mathcal{N}(A) =: N$ as a $(4n \times (n - r))$ matrix, we have arrived at a set of $n - r$ free parameters $\omega$ that we can optimize with respect to. To recover the full set of coefficients $\xi$, we can use the following equation:

$$\xi = N\omega \tag{34}$$

and evaluate the spline at the desired points to get the curve $\gamma(t)$.

### C.2. Optimizing the spline to find a geodesic

In the previous subsection, we have reduced the infinite set of functions in which we are looking for a geodesic to another but considerably smaller, infinite set of splines. The next step is to use optimization to find the spline that minimizes the energy defined in Definition A.21. Calculating the energy requires computing an integral, which is, in practice, approximated by a sum over a discretized interval.

In the following treatment we assume that $\mathcal{D} = \mathbb{R}^n$ and let $f_\theta : \mathcal{Z} \to \mathbb{R}^n$ be a decoder parametrized by $\theta$ and $\gamma : [0, 1] \to \mathcal{Z}$ be a spline in the latent space, then the approximation of the energy of $\gamma$ is given by:

$$\begin{aligned} E(\gamma) &= \frac{1}{2} \int_0^1 |\gamma'(t)|_g^2 dt \\ &= \frac{1}{2} \int_0^1 |\frac{\partial}{\partial t} f_\theta(\gamma(t))|_E^2 dt \\ &\approx \frac{1}{2\Delta t} \sum_{i=2}^{n_t} \|f_\theta(\gamma(\bar{t}_i)) - f_\theta(\gamma(\bar{t}_{i-1}))\|^2 =: \bar{E}(\gamma) \end{aligned} \tag{35}$$

where $\{\bar{t}_i\}_{i=0}^{n_t}$ is a sequence of $n_t$ points in the interval $[0, 1]$. Combining this with the discussion in the previous section, we can now define the optimization problem as simply:

$$\min_{\boldsymbol{\omega}} \quad \bar{E}(\gamma_{\boldsymbol{\omega}}) \tag{36}$$

where we use $\boldsymbol{\omega} = \{\omega_j\}_{j=1}^k$ to de note the parameters of the $k$ different splines given the dimensionality of the latent space $\mathcal{Z} \in \mathbb{R}^k$ and remind the reader that $\gamma_{\boldsymbol{\omega}}(t) = (\gamma_{\omega_1}^1(t), \ldots, \gamma_{\omega_k}^k(t)) \in \mathcal{Z}$.

Using optimization to learn the manifold will result in different approximations depending on the initialization of the parameters and the optimization algorithm used. Considering the problem in light of the first line of Eq. 35, we can see that the Riemannian metric becomes the stochastic term. In this sense, the manifold is stochastic, and the resulting distances between points will be affected by this stochasticity.

Following (Syrota et al., 2024), having access to an ensemble of decoders allows us, in principle, to make the optimization problem in Eq. 36 aware of the uncertainty involved. The methodology effectively uses Monte Carlo methods to compute the energy with respect to the uncorrelated posterior over parameters. This posterior approximated by an ensemble. The following equation is the optimization problem we solve and makes the idea explicit:

$$\min_{\boldsymbol{\omega}} \quad \frac{1}{2\Delta t} \sum_{i=2}^{n_t} \left\| f_{\hat{\theta}_j}(\gamma_{\boldsymbol{\omega}}(\bar{t}_i)) - f_{\hat{\theta}_k}(\gamma_{\boldsymbol{\omega}}(\bar{t}_{i-1})) \right\|^2 \tag{37}$$

where $\Delta_t = \bar{t}_i - \bar{t}_{i-1}$ is the step size in the discretization of the interval $[0, 1]$ and is assumed to be constant. The decoders $f_{\hat{\theta}_k}$ and $f_{\hat{\theta}_{kj}}$ are sampled uniformly and independently from the ensemble.

# D. Experiments

### $\mathcal{M}$-flow on MNIST and CIFAR10

The decoder is modeled by 10 coupling flow layers introduced by (Dinh et al., 2015) where the conditioner is 2-layer MLP with 50 hidden units and ReLu activation function, and transformer is the RQS spline (Durkan et al., 2019) in the interval $[-3, 3]$ with 8 knots (22 knots for CIFAR10). The decoder has 5.5 million parameter for MNIST and 55 million for the CIFAR10 model.

The encoder consists of 4 Residual Blocks (He et al., 2015) with Convolutional transformations each time multiplying the number of channels by 5 and the Relu activation function. The last layer is the linear layer with hyperbolic tangent function multiplied by 3 at the end to constrain the output to [-3,3] and accommodate the coupling flow above. The encoder has 135 thousand parameters for both MNIST and CIFAR10 models.

Adam (Kingma & Ba, 2017) was used for all the experiments.

### VAE on FMNIST and CELEBA

The encoder consists of five $4 \times 4$ convolutional layers, the first two with 128 channels, followed by the next three with 256 channels, with the first and last having a stride of one and the others a stride of two; two linear layers, one with 256 units and the second with 64 units (latent dimensions) for the VAE approximate posterior mean and 64 for the VAE approximate posterior variance.

The decoder starts with two fully-connected layers, one with 256 units and one with 16384 units; five $4 \times 4$ resize convolutional layers (Odena et al., 2016), with the same stride configuration as the encoder, but with 256, 128, 64, and 32 channels, respectively. All layers in the encoder and decoder have an ELU activation function. Total number of parameters: 6.5 million.

The VAE is trained with an additional loss term coming from the Perceptual loss (Johnson et al., 2016).

### Geodesic computation

The geodesic computation was done on models that had an ensemble of 8 decoders following the procedure of (Syrota et al., 2024) and hyperparameters in Table 2.

| Parameter | Value |
|---|---|
| Initialization of free parameters | zeros (straight line) |
| Number of polynomials in the spline | 10 |
| Discretization in time (energy) | 256 |
| Discretization in time (final length) | 256 |
| Optimizer | Adam (Kingma & Ba, 2017) |
| Max. steps | 4096 |
| Learning rate | 0.01 |
| Early stopping patience (steps) | 100 |
| Early stopping delta | 1.0 |

Table 2: Shared geodesics training parameters for all models.

