# OpenReview forum: "Identifying Metric Structures of Deep Latent Variable Models"
_ICML.cc/2025/Conference — ICML 2025 poster_

### Official Review · Reviewer_ev8z · 2025-02-28

**Overall Recommendation:** 4

**Summary:**

This paper addresses the problem of learning identifiable representations from a novel perspective, focusing on the distances between representations rather than their coordinates. The authors begin by discussing the challenge of identifiability in latent variable models, emphasizing that maximum likelihood estimation (MLE) alone does not guarantee identifiability. They highlight that training the same model twice can result in different learned representations. To address this, the paper introduces a new notion of identifiability: given two models, A and B, the geodesic distance between latent variables $z_1$ and $z_2$ is considered identifiable if it remains consistent across both models. To validate this idea, the authors train multiple VAEs on different datasets and compare Euclidean and geodesic distances between 100 randomly selected test sample pairs. Their results show that geodesic distances exhibit lower variance, supporting their proposed notion of identifiability.

**Claims And Evidence:**

See Weaknesses

**Essential References Not Discussed:**

See Relation To Broader Scientific Literature*

[1] Chadebec, Clément, and Stéphanie Allassonnière. "A geometric perspective on variational autoencoders." Advances in Neural Information Processing Systems 35 (2022): 19618-19630.

[2] Chen, Nutan, et al. "Fast approximate geodesics for deep generative models." Artificial Neural Networks and Machine Learning–ICANN 2019: Deep Learning: 28th International Conference on Artificial Neural Networks, Munich, Germany, September 17–19, 2019, Proceedings, Part II 28. Springer International Publishing, 2019.

[3] Chen, Nutan, et al. "Metrics for deep generative models." International Conference on Artificial Intelligence and Statistics. PMLR, 2018.

[4] Arvanitidis, Georgios, Lars Kai Hansen, and Søren Hauberg. "Latent space oddity: on the curvature of deep generative models." arXiv preprint arXiv:1710.11379 (2017).

[5] Higgins, Irina, et al. "Towards a definition of disentangled representations." arXiv preprint arXiv:1812.02230 (2018).

[6] Zhu, Xinqi, Chang Xu, and Dacheng Tao. "Commutative lie group vae for disentanglement learning." International Conference on Machine Learning. PMLR, 2021.

[7] Wang, Tan, et al. "Self-supervised learning disentangled group representation as feature." Advances in Neural Information Processing Systems 34 (2021): 18225-18240.

[8] Yang, Tao, et al. "Towards building a group-based unsupervised representation disentanglement framework." arXiv preprint arXiv:2102.10303 (2021).

**Experimental Designs Or Analyses:**

See Weaknesses

**Methods And Evaluation Criteria:**

See Weaknesses

**Other Comments Or Suggestions:**

- I want to applaud the authors for writing Section 7.

**Other Strengths And Weaknesses:**

**Strengths**:

- The paper reads really well. And it offers a good introduction to the identifiability problem

- The key contributions of the paper namely Theorem 4.7 is a good theoretical contribution and worth highlighting.

- Regardless of the results: the experiment design is very systematic and justified.

**Weaknesses**

- The theoretical framework, to the best of my ability, checks out. However, I find that the experiments do not strongly validate the hypothesis that geodesic distances are truly identifiable. The main result, particularly Figure 7, is somewhat underwhelming, as there is considerable overlap between the two histograms. While this does suggest that geodesic distances are more identifiable than Euclidean ones, I would argue that this falls short of demonstrating that they are identifiable in a definitive sense. Ofcourse, there are some notable challenges in measuring geodesic distances accurately. Specifically: (1) 1. The optimization procedure used to compute geodesic distances is not optimal, as the authors themselves acknowledge. Euclidean distance in the data space is not an ideal metric for these datasets. A useful addition to the paper would be a toy experiment where the data manifold is known, allowing for precise measurement of the true geodesic distances.

- As pointed out in the prior work section, while this is the 1st work to the best of my knowledge that points out the connection between identifiability and transformation, there is a strong link between this work and the literature on lie group latent spaces [5-8]. The authors need to discuss some of these works and tone down the claim that this is the first work that focused on transformations.

- While the authors point this out in the first paragraph of Section 7, it remains a strong problem. The claim "We argue that most data is equipped with units of measurement, which greatly simplifies the task of picking a suitable metric in the observation space." does not hold in most vision and nlp tasks for example. Moreover, if we're using Euclidian distance in x-space, we implicitly assume that the dataset is dense, which is not a very reliable assumption.


Overall, I think the connection between identifiability and distances is a perfectly valid contribution worthy of publishing and of interest to the representation learning community. The main weaknesses of the paper in my opinion, are: (1) Not discussing the connection between this work and all the disentanglement work that was built on [4] using groups ("transformations") (2) The results sadly are not very promising.

**Questions For Authors:**

- In Figure 1: I'm wondering if you can show the the matrices using the geodesics distances as well instead of Euclidean? How would the matrices for different runs look like?

**Relation To Broader Scientific Literature:**

- I think the paper misses an important section of literature: Disentangled representations + lie groups, namely [5] and all the papers that were built on top of that (e.g. [6-8]) as well as many other equivariant neural networks. Currently, the paper reads in a way that is the first paper that thought about representations in terms of transformations. This claim needs to be toned down quite a bit I would say. As these papers show the group element $g$ $g. x_1 = x_2$  indeed corresponds to the geodesics.

- Furthermore, while I understand that the authors do not make a big claim on proposing a new way to compute geodesics. However previous works on computing and analyzing geodesics must be cited: (3.g. [1-4])

**Theoretical Claims:**

The theoretical framework, to the best of my ability, checks out.

---

> ### Author Rebuttal · Authors · 2025-04-01
>
> We thank the reviewer for thoughtful feedback, as well as their support for acceptance.
>
> >However, I find that the experiments do not strongly validate the hypothesis that geodesic distances are truly identifiable. The main result, particularly Figure 7
>
> We emphasize that the main result of the paper is Theorem 4.5. It shows that under any (injective) generative model, the indeterminacy transformations of the true latent space will automatically respect the true geometry of the data. I.e. it proves that any geometrical information extracted from that model is identifiable. We achieve this without placing (notable) restrictions on the model, architecture, or training methods.
>
> We motivate our paper and focus our experiments within the point of view of geodesic distances which is an example of geometrical information used in practice. This leads to Theorem 4.7 proving that these are identifiable.
>
> >there is considerable overlap between the two histograms. [..] I would argue that this falls short of demonstrating that they are identifiable in a definitive sense
>
> We want to clarify that identifiability is a theoretical question and that our present theorems are the definite evidence.
>
> The experiments demonstrate that the asymptotic property in Theorem 4.7 is practically achievable in standard models using off-the-shelf methods on finite data.
>
> However, we understand the concern about overlapping histograms, but emphasize that such is to be expected and is fully in line with the theory:
> - Proposition 4.8 shows that a (scaled) Euclidean distance can be identifiable if the model behaves in a Euclidean way in a region.
> - Riemannian geometry is locally Euclidean, implying that local Euclidean distances can be expected to be robust when points are close.
> Consequently, neighboring points can have robust Euclidean distances (low CoV), implying overlapping histograms.
> Finite data further implies uncertainty compared to theoretical treatment and makes the estimation of the manifold stochastic. As the reviewer points out, optimization of geodesics is noisy and may lead to a distorted picture (no efforts were made to counteract this on specific datasets or models).
>
> While the focus of our work is theoretical, we acknowledge the reviewers’ requests for additional experiments, and have added results for FMNIST and CIFAR10. See [link](https://tinyurl.com/yxzwks2j) for plots and tables. FMNIST results are similar to previous results, while CIFAR10 shows the clear separation of histograms that you have requested.
>
> >there is a strong link between this work and the literature on lie group latent spaces [5-8]
>
> We do not claim to be the first to explore transformations of latent space; however, we are the first to apply them in the context of identifiability using Riemannian geometry.
>
> While previously used for different purposes, latent space transformations have proven valuable in various areas, including disentangled and equivariant learning, highlighting their mathematical and conceptual connections.
> The mentioned literature assumes a disentangled latent space where transformations decompose into individual factors of variation and the goal is to find representations that respect this structure (roughly speaking, equivariance of $A_{a,b}$ (Def 4.2)). Instead of enforcing specific properties, our theory analyzes the natural properties of $A_{a,b}$, and we find that just by learning a generative model, $A_{a,b}$ will automatically respect the latent Riemannian geometry (Theorem 4.5).  Thus, our theory is valid regardless of whether a disentangled latent space exists in the sense of [5].
>
> >previous works on computing and analyzing geodesics must be cited
>
> We already cite [4] and will include [1-3] as appropriate.
>
> >The claim "[...] most data is equipped with units of measurement[...]." does not hold in most vision and nlp tasks
>
> We acknowledge that there are cases where picking a suitable metric in data space is not trivial, but emphasize that this metric should only be meaningful infinitesimally. E.g. Euclidean distances are generally unsuited for images, but infinitesimally they are perfectly reasonable. Our theory also applies when pulling back ‘perceptual distances’, e.g. using features from pre-trained neural networks, see e.g. [this paper](https://tinyurl.com/3jvx27r3).
>
> >using Euclidian distance in x-space, we implicitly assume that the dataset is dense
>
> We disagree with this statement as we are not measuring 'isomap-style' geodesics. We measure geodesics along the manifold spanned by the model which does not require dense data to be identifiable.
>
> >The main weaknesses of the paper are: (1) Not discussing the connection between this work and all the disentanglement work (2) The results sadly are not very promising.
>
> To conclude:
> 1. see the discussion above,
> 2. our main results are theoretical, and the experiments are fully consistent with the theory. Our released code provides a path for turning theory into practice.

---

> > ### Comment · Reviewer_ev8z · 2025-04-02
> >
> > I have read the rebuttal. I want to thank the authors for their response.
> >
> > Just to double-check, in Figure 7, if we had measured the geodesic distance exactly, the histogram should fully peak at 0 correct? (in the mnist case at least). I guess still not, given that the 30 models trained with different seeds don't all have exactly the same likelihood? Is there anything else in the theory that breaks here?
> >
> > > We do not claim to be the first to explore transformations of latent space; however, we are the first to apply them in the context of identifiability using Riemannian geometry. While previously used for different purposes, latent space transformations have proven valuable in various areas, including disentangled and equivariant learning, highlighting their mathematical and conceptual connections. The mentioned literature assumes a disentangled latent space where transformations decompose into individual factors of variation and the goal is to find representations that respect this structure
> >
> > This makes sense. I would add a version of this in the final version.
> >
> > Given that I think I underestimated the importance of 4.5, I will increase my score.

---

> > > ### Author Response · Authors · 2025-04-03
> > >
> > > We thank the reviewer for the increased score and follow-up clarifications.
> > >
> > > >Just to double-check, in Figure 7, if we had measured the geodesic distance exactly, the histogram should fully peak at 0 correct? (in the mnist case at least). I guess still not, given that the 30 models trained with different seeds don't all have exactly the same likelihood? Is there anything else in the theory that breaks here?
> > >
> > > We share your intuition. Having exact geodesic distances would result in the histogram shifting closer to 0. However, there is still the noise associated with finite data which makes the manifold stochastic and hence we cannot expect no variability at all.
> > >
> > > > This makes sense. I would add a version of this in the final version.
> > >
> > > We will update the final version putting more emphasis on how our contributions relate to the literature in the field.

---

### Official Review · Reviewer_TdYi · 2025-03-14

**Overall Recommendation:** 3

**Summary:**

In this paper, the authors address the challenge of statistical identifiability in deep latent variable models, which are used to extract condensed representations of data. Traditional methods attempt to improve identifiability by imposing constraints like labeled data or limiting model expressivity. Instead, the authors shift the focus from identifying individual latent variables to identifying meaningful relationships between them—such as distances, angles, and volumes.
The authors prove that these geometric relationships can be statistically identified under minimal assumptions, without additional labeled data. This result is significant for fields like scientific discovery, where reliable data interpretation is crucial. In the experiments, the authors test their assumption over 2 different datasets, MNIST and Celeb-A. They performed Student's T-test to show that the geodesic distances have much less variance than Euclidean distance.

**Claims And Evidence:**

I believe all the claims are well supported.

**Essential References Not Discussed:**

I think the literature is reviewed well.

**Ethical Review Concerns:**

There is no ethical concern.

**Experimental Designs Or Analyses:**

I find the claim in the result analysis confusing. For example,
1. Why do the authors only use 3 classes of MNIST? Is this cherry-picked?
2. I don't see why we should expect that the digit class 0, 5, and 7 are naturally close to each other. I hope the authors can explain it in more details.

**Methods And Evaluation Criteria:**

I believe there can be more benchmarks included in this paper. Other commonly used image datasets, such as Fashion-MNIST, SVHN, Cifar-10, should also be computationally cheap to run.

**Other Comments Or Suggestions:**

I don't have any other comments.

**Other Strengths And Weaknesses:**

The goal of this work is well motivated and the paper is well strucutured in general.

**Questions For Authors:**

I have listed my questions in previous sections.

**Relation To Broader Scientific Literature:**

A key contribution of the paper is linking identifiability to Riemannian geometry, establishing a novel theoretical framework. This connection allows practitioners to leverage established Riemannian tools (e.g., Riemannian averages, covariances, and principal components) to analyze latent structures in a statistically sound manner.

**Theoretical Claims:**

I believe the theoretical claims are sound.

---

> ### Author Rebuttal · Authors · 2025-04-01
>
> We sincerely appreciate the reviewer's valuable feedback and support for acceptance.
>
> >I believe there can be more benchmarks included in this paper. Other commonly used image datasets, such as Fashion-MNIST, SVHN, Cifar-10, should also be computationally cheap to run.
>
> While the focus of our work is theoretical (see reply to reviewer ev8z for more details), we acknowledge the reviewers’ requests for additional experiments.
>
> In particular, addressing this review we run experiments of FMNIST and CIFAR10. Following the approach in the main paper, we split them into an experiment satisfying the injectivity constraint (CIFAR10) and an experiment that does not satisfy the constraint (FMNIST). The histograms and an updated table are [at this link](https://tinyurl.com/yxzwks2j) and we provide the table here as well.
> |                | MNIST  | CELEBA  | FMNIST | CIFAR10 |
> |-------------- |--------|--------|--------|--------|
> | **t-statistic** | -8.64  | -22.33 | -16.75 | -42.83 |
> | **p-value**    | 1.00   | 1.00   | 1.00   | 1.00   |
>
> *Table: One-sided Student's t-test for the variability of geodesic versus Euclidean distances.*
>
> The findings are similar to those reported in the submitted paper, which supports the presented theory.
>
> We will further extend the paper with an appendix including extra results and details on implementations and model choices.
>
> >1. Why do the authors only use 3 classes of MNIST? Is this cherry-picked?
>
> As mentioned in the submitted paper, the choice of 3 classes for MNIST was to simplify plotting. We point to the extra experiments above and code in the submitted supplementary material (CelebA) to document absence of any cherry-picking.
>
> >2. I don't see why we should expect that the digit class 0, 5, and 7 are naturally close to each other. I hope the authors can explain it in more details.
>
> We do not expect any classes to be naturally close to each other. The digits 0,5 and 7 were picked randomly and their placement in Fig. 5 is merely a consequence of optimization of the model fitting.
>
> We hope to have addressed your concerns and sincerely thank you again for your review.

---

### Official Review · Reviewer_Rhp6 · 2025-03-18

**Overall Recommendation:** 3

**Summary:**

This paper studies the geometry of latent spaces of latent variable models like VAEs, normalizing flows, diffusion models etc. Primarily, the authors highlight that many seemingly simple factors like the latent coordinates or their pairwise euclidean distances etc. of latent variable models are provably not identifiable. This means that under a probablistic setting: despite having different parameterizations- they do not lead to the same distributions. The motivation of the generative model is to discover meaningful intrinsic properties of data that should not be dependent on randomness that is inherent due to training, noise etc.

The authors explore this in the context of differential geometry of the latent space of generative models. The main presmise is that unlike the latent coordinates or even euclidean distances between them -  geodesic distances computed using a pullback metric from the observation space satisfies identifiability. This paper both rigorously proves this and emperically demonstrates the hypothesis with experiments from MNIST and and CELEBA.

**Claims And Evidence:**

Yes. The central point is proved and also demonstrated emperially. However - I do find the experiments lacking generality in the class of generative models investigated.

**Essential References Not Discussed:**

Its fine

**Experimental Designs Or Analyses:**

There is scope to be much more comprehensive in the experiments - for e.g. I would be very glad to see a table similar to Figure 7 for the transcriptomic data example from Figure 1 and that would solidify the main message of the paper across different models and types of data.

**Methods And Evaluation Criteria:**

Yes.

**Other Comments Or Suggestions:**

I am tempted to make the conclusion that generative models when trained well, tend to produce representations that preserve "intrinsic" distances on the data-manifold. For example - If I leave out the decoder 'f' completely and simply use my training dataset with a k-nearest neighbor graph and then compute a Dijkstra-like shortest path distance on this graph - I suspect it would correlate quite well with the construction of the pullback metric and computation of the geodesic distance from (35) which very much depends on the chosen model 'f'. It would be nice to have some experiments where the variability in the geodesic is also visualized somehow (like Fig 5 but with a band instead of just one curve) - especially in comparison to this model-independent geodesic distance.

**Other Strengths And Weaknesses:**

Overall - I think this is a nice paper with a comprehensive conceptual and theoretical treatise on developing parameterization invariant representations. However, the experiments do lack generality and some more convincing demonstrations of the core message would go a long way. Therefore, I am very much on the border leaning slightly positively because of a well-compiled submission and an interesting read.

**Questions For Authors:**

- Is Table 1 reporting the values for the geodesic or Euclidean distance? I am unable to parse the message here
- Is the familiarity in the trajectories reported in Figure 6 always the case? I was expecting that the trajectory of the Euclidean geodesic to give images with comparatively more "abrupt" changes in each step in comparison to the geodesic which should give a more seamless transformation. It is somewhat visible already - although not strikingly.

**Relation To Broader Scientific Literature:**

To the best of my knowledge - this paper discusses an important issue that even though is not unique in all literature - provides a novel comprehensive analysis: theoretically and empirically regarding parameterization invariance of generative models.

**Theoretical Claims:**

Yes

---

> ### Author Rebuttal · Authors · 2025-04-01
>
> We are grateful to the reviewer for valuable feedback and favoring acceptance.
>
> >There is scope to be much more comprehensive in the experiments - for e.g. I would be very glad to see a table similar to Figure 7 for the transcriptomic data example from Figure 1 and that would solidify the main message of the paper across different models and types of data.
>
> While the focus of our work is theoretical (see reply to reviewer ev8z for more details), we acknowledge the reviewers’ requests for additional experiments and will extend the paper with an Appendix including extra results and details on implementations and model choices.
>
> In particular, addressing this point we run experiments of FMNIST and CIFAR10. Following the approach in the main paper, we split them into an experiment satisfying the injectivity constraint (CIFAR10) and an experiment that does not satisfy the constraint (FMNIST). The histograms and an updated table are available [at this link](https://tinyurl.com/yxzwks2j). The findings are similar to those reported in the submitted paper, which supports the presented theory.
> Furthermore, we share the sentiment that the transcriptomic data example is underexplored and plan to add a distance matrix for the geodesic distances to Figure 1.
>
> >I am tempted to make the conclusion that generative models when trained well, tend to produce representations that preserve "intrinsic" distances on the data-manifold.
>
> Indeed, the main result of the paper, Theorem 4.5, shows that under any well-trained (injective) generative model, the indeterminacy transformations of the true latent space will automatically respect the true geometry of the data. I.e. it proves that any geometrical information extracted from the model is identifiable.
>
> >For example - If I leave out the decoder 'f' completely and simply use my training dataset with a k-nearest neighbor graph and then compute a Dijkstra-like shortest path distance on this graph - I suspect it would correlate quite well with the construction of the pullback metric and computation of the geodesic distance from (35) which very much depends on the chosen model 'f'.
>
> We appreciate and share your intuition. [This paper](https://arxiv.org/abs/1812.08284) considers geodesics under such an approach. However, we should emphasize that this is heuristic and is not strictly tied to our theoretical results on identifiability. Practically, we expect the approach to work well, though.
>
> >It would be nice to have some experiments where the variability in the geodesic is also visualized somehow (like Fig 5 but with a band instead of just one curve) - especially in comparison to this model-independent geodesic distance
>
> We agree that different optimisations can lead to different geodesics between the same two points, and acknowledging that a geodesic in itself is not unique. However, in this paper we address the variability of the distance measure across retraining of the models themselves. In our experiments, this means that we compute the geodesic distance between the same two points across 30 different models with 30 different latent spaces. Fig.5 shows just one geodesic in one latent space. Therefore plotting the band would be concerned with a different kind of variability.
>
> > Is Table 1 reporting the values for the geodesic or Euclidean distance? I am unable to parse the message here
>
> Building on the above, for each pair of points we have 30 measurements of distances according to both Euclidean and geodesic measures. To demonstrate that the geodesic distance is more stable, we compute the coefficient of variation (CV) of both distance measures for each point pair. These are then plotted in Fig. 7 and Table 1 reports the Student’s t-test for the CVs with one sided null hypothesis that geodesics are more stable, i.e. has a smaller CV. The message of Table 1 is that geodesic distances exhibit significantly less variation than Euclidean distances.
>
> We acknowledge that the table caption should be improved. We will make these details more explicit and will further add an appendix detailing the experiments.
>
> > Is the familiarity in the trajectories reported in Figure 6 always the case? I was expecting that the trajectory of the Euclidean geodesic to give images with comparatively more "abrupt" changes in each step in comparison to the geodesic which should give a more seamless transformation. It is somewhat visible already - although not strikingly.
>
> Your expectation is correct. We often see that geodesics provide more ‘smooth’ interpolations while Euclidean interpolations are more ‘abrupt’. This happens because geodesics move with constant speed in data space. This trend is, however, more evident, e.g., in MNIST than in CelebA. We’re happy to include such MNIST examples in the appendix if they are deemed interesting.
>
> We hope that our clarifications and additional experiments strengthen your view of the paper, and we thank the reviewer for the thoughtful feedback.

---

### Decision · Program_Chairs · 2025-05-01

**Decision:**

Accept (poster)

**Comment:**

This paper investigates identifiability in latent variable models, focusing on identifiability of geometric features such as distances in the latent space. The core of the paper are theoretical results, on the identifiability of a Riemannian metric in the latent space of a deep generative model, and of related Riemannian geometry properties. Reviewers would have liked to see more experimental validation of the learned geodesics, specifically as there is still some unexpected variation in Figure 7, but all reviewers are in favour of accepting the paper, in particular given the potential importance of the theoretical results.